# Learning Posterior Predictive Distributions for Node Classification from Synthetic Graph Priors

**Jeongwhan Choi**[*][✉], **Jongwoo Kim,**[*]  **Woosung Kang,**   **Noseong Park**[†]
KAIST
Daejeon, Republic of Korea

## Abstract

One of the most challenging problems in graph machine learning is generalizing across graphs with diverse properties. Graph neural networks (GNNs) face a fundamental limitation: they require separate training for each new graph, preventing universal generalization across diverse graph datasets. A critical challenge facing GNNs lies in their reliance on labeled training data for each individual graph, a requirement that hinders the capacity for universal node classification due to the heterogeneity inherent in graphs — differences in homophily levels, community structures, and feature distributions across datasets. Inspired by the success of large language models (LLMs) that achieve in-context learning through massive-scale pre-training on diverse datasets, we introduce NodePFN. This universal node classification method generalizes to arbitrary graphs without graph-specific training. NodePFN learns posterior predictive distributions (PPDs) by training only on thousands of synthetic graphs generated from carefully designed priors. Our synthetic graph generation covers real-world graphs through the use of random networks with controllable homophily levels and structural causal models for complex feature-label relationships. We develop a dual-branch architecture combining context-query attention mechanisms with local message passing to enable graph-aware in-context learning. Extensive evaluation on 23 benchmarks demonstrates that a single pre-trained NodePFN achieves 71.27% average accuracy. These results validate that universal graph learning patterns can be effectively learned from synthetic priors, establishing a new paradigm for generalization in node classification.

## 1 Introduction

Graph neural networks (GNNs) have achieved success in tasks on graph-structured data prevalent in chemistry (Gilmer et al., 2017; Hamilton, 2020), recommender systems (Ying et al., 2018; He et al., 2020), biology (Bongini et al., 2022), social sciences (Kipf & Welling, 2017; Qiu et al., 2018), etc, by learning to aggregate neighborhood information through message passing. However, GNNs still have the limitation that, for node classification (Kipf & Welling, 2017; Bresson & Laurent, 2017; Hamilton et al., 2017; Klicpera et al., 2019; Zhou et al., 2020; Choi et al., 2023; Luan et al., 2023; Kim et al., 2025; Choi et al., 2025b), separate GNN models must be trained for the labeled nodes of each new graph. This dependence on graph-specific training makes generalization across graphs with different properties challenging. The core issue is that real-world graphs exhibit vastly different structural properties — varying homophily levels, community structures and features, and degree distributions among datasets. GNNs struggle to handle this diversity without dataset-specific training.

The success of foundation models, particularly large language models (LLMs) (Brown et al., 2020; Touvron et al., 2023; Achiam et al., 2023) comes from their training paradigm of learning generalizable patterns from massive and diverse datasets. This enables these models to perform in-context learning, adapting to new datasets without parameter updates by learned patterns during pre-training.

---

[*]Equal contribution.
[†]Corresponding Author.

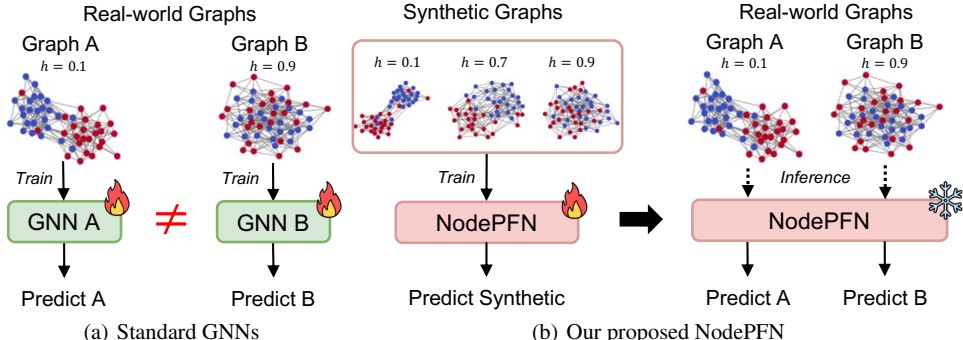

Figure 1: NodePFN enables universal node classification. (a) Each real-world graph requires its own trained GNN model. (b) Pre-training on synthetic graphs sampled from controlled priors (varying homophily ($h$) from 0.1 to 0.9) produces *a single model* capable of generalization to arbitrary graphs.

In a manner analogous to the capacity of LLMs to adapt to new samples with only context examples, we propose a graph model that performs node classification on arbitrary graph datasets. This implies that *a single pre-trained model* could perform node classification on arbitrary graph datasets without needing to be trained specifically for that dataset.

Recent studies have explored applying LLMs to graphs (Li et al., 2024a; Chen et al., 2024b;a; Li et al., 2024c; Tang et al., 2024; Liu et al., 2024). However, LLMs, primarily trained on textual data, are better suited for capturing semantic content rather than learning the structural patterns that govern node classification on diverse graph topologies. While Zhao et al. (2025) introduce a fully inductive framework, it still requires training on specific source datasets, with performance varying significantly based on the training dataset choice.

We propose a different approach by training on synthetic graphs that systematically cover the diversity of real-world graphs (see Fig. 1). The key insight is learning the posterior predictive distribution (PPD) from synthetic priors. Recently, prior-fitted networks (PFNs) have demonstrated that models trained on synthetic data from carefully designed priors can approximate PPDs for new datasets in a single forward pass (Müller et al., 2022; Hollmann et al., 2023). This approach enables in-context learning. That is, the model learns to extract patterns from context examples (labeled nodes) and apply them to query points (unlabeled nodes), enabling immediate prediction without gradient updates. We extend this PFN paradigm to graphs by designing synthetic graph priors that systematically control homophily levels, community structures, and feature-label relationships. We aim to design a model that predicts the label distribution of query nodes based on labeled context nodes in real graphs, by learning PPD from various synthetic graphs.

We introduce **NodePFN** learning PPDs for node classification from synthetic graph priors. During training, we generate thousands of diverse synthetic graphs, leveraging methods that control class homophily and community levels to ensure that they include a range of network characteristics found in real-world benchmarks.

Experimental evaluation on 23 real-world benchmarks shows that NodePFN achieves competitive performance. Our approach outperforms on both homophily and heterophily graph benchmarks, surpassing GNN baselines. These extensive experiments validate that the patterns governing node classification can *de facto* be learned from synthetic priors.

The contributions of our proposed NodePFN[1] are summarized as follows.

- To the best of our knowledge, we are the first to extend the PFN paradigm to graphs, demonstrating that PPDs for node classification can be learned from synthetic graph prior distributions without requiring actual training data. (Section 3).
- We design a comprehensive synthetic graph prior by using random networks, incorporating levels of homophily, community structure, and feature-label relationships. (Section 3.2).

---

[1]Our code is available here: `https://github.com/jeongwhanchoi/NodePFN`

- To enable learning graph-aware context from both labeled examples and topological structure, we developed a novel dual-branch architecture combining a context-query attention mechanism with local message passing (Section 3.3).
- We demonstrate universal node classification across 23 diverse real-world benchmarks using a single pre-trained model, achieving an average accuracy of 71.27% and strong performance of 65.14% on challenging heterophily graphs where traditional GNNs struggle (Section 4).

## 2 PRELIMINARIES

In this section, we introduce posterior predictive distribution and prior-data fitted networks. Then, we address the notation used in our study and node classification

### 2.1 POSTERIOR PREDICTIVE DISTRIBUTION IN SUPERVISED LEARNING

In supervised learning, the goal is to predict labels for unlabeled data points using labeled training samples. Given a training set $\mathcal{D}_{\text{train}} = \{(\mathbf{x}_i, \mathbf{y}_i)\}_{i=1}^n$ and test set $\mathcal{D}_{\text{test}} = \{\mathbf{x}_j\}_{j=1}^m$, we aim to predict labels for the test set. In the Bayesian framework, we model the conditional distribution $p(\mathbf{y}|\mathbf{x}; \phi)$ with parameters $\phi$ treated as random variables with prior $p(\phi)$. The goal is to predict labels for a test point $\mathbf{x}_{\text{test}}$ using the posterior predictive distribution (PPD):

$$p(\mathbf{y}_{\text{test}}|\mathbf{x}_{\text{test}}, \mathcal{D}_{\text{train}}) = \int p(\mathbf{y}_{\text{test}}|\mathbf{x}_{\text{test}}, \phi)p(\phi|\mathcal{D}_{\text{train}})d\phi, \tag{1}$$

where the posterior distribution follows Bayes' rule:

$$p(\phi|\mathcal{D}_{\text{train}}) \propto p(\phi) \prod_{i=1}^n p(\mathbf{y}_i|\mathbf{x}_i; \phi). \tag{2}$$

If the hypothesis class includes the true conditional distribution, there exists a $\phi^*$ such that $p(\mathbf{y}|\mathbf{x}; \phi^*) = p_{\text{true}}(\mathbf{y}|\mathbf{x})$ for all $(\mathbf{x}, \mathbf{y})$, then the PPD results in optimal prediction.

### 2.2 PRIOR-DATA FITTED NETWORKS

Prior-data fitted networks (PFNs) (Müller et al., 2022) learn an approximation of the PPD from the training data using neural networks. Instead of computing the integral in Eq. (1) at test time, PFNs are trained on synthetic datasets sampled from a prior $p(\mathcal{D})$ to learn:

$$f_\theta : (\mathbf{x}_{\text{test}}, \mathcal{D}_{\text{train}}) \mapsto p(\mathbf{y}_{\text{test}}|\mathbf{x}_{\text{test}}, \mathcal{D}_{\text{train}}). \tag{3}$$

During training, we sample synthetic datasets from a prior $p(\mathcal{D})$. Each dataset is split into training and test sets. The PFN $f_\theta$ with parameters $\theta$ is trained to minimize the expected loss:

$$\mathcal{L}(\theta) = \mathbb{E}_{\mathcal{D} \sim p(\mathcal{D})} \left[ -\log q_\theta(\mathbf{y}_{\text{test}}|\mathbf{x}_{\text{test}}, \mathcal{D}_{\text{train}}) \right], \tag{4}$$

where $q_\theta$ is the neural network's approximation of the true PPD. By training on synthetic datasets, the model learns to extract relevant patterns from context samples and apply them to new queries.

This approach allows the model to perform inference at test time in a single forward pass without gradient updates, given a new dataset. Through implicit Bayesian inference, the network learns to marginalize over parameter uncertainty.

### 2.3 GNNs FOR NODE CLASSIFICATION AND THEIR LIMITATIONS

In the node classification problem, given a graph $\mathcal{G} = (\mathcal{V}, \mathcal{E})$ with node feature matrix $\mathbf{X} \in \mathbb{R}^{|\mathcal{V}| \times d}$ where $d$ is the feature dimension, adjacency matrix $\mathbf{A} \in \{0, 1\}^{|\mathcal{V}| \times |\mathcal{V}|}$, and a set of labeled nodes $\mathcal{V}_{\text{train}} \subset \mathcal{V}$ with their corresponding labels $\mathbf{y}_{\text{test}}$, the goal is to predict labels $\hat{\mathbf{y}}_{\text{test}}$ for the unlabeled node set $\mathcal{V}_{\text{test}} = \mathcal{V} \setminus \mathcal{V}_{\text{train}}$.

**Homophily in Node Classification.** The success of GNNs is believed to be rooted in the homophily assumption (McPherson et al., 2001), which implies that connected nodes tend to share similar attributes (Hamilton, 2020). This provides additional useful information in the aggregated features compared to the original node features, and the effectiveness of node classification can be determined by the level of edge homophily (Luan et al., 2023; Zhu et al., 2020a), which measures the tendency of connected nodes to share the same class label. The level of homophily, $h$, falls within the range of $[0, 1]$, with a value closer to 1, strong homophily, implying that GNNs are more likely to outperform than non-graph models, and vice versa.

## 2.4 RANDOM GRAPH MODELS

We consider 2 random networks. Erdős-Rényi (ER) model (Erdős & Rényi, 1959) generates graphs where each edge appears independently with probability $p_{\text{er}}$. This creates graphs with binomial degree distributions and no inherent community structure. Stochastic block models (SBMs) (Holland et al., 1983) control community structure through different connection probabilities within and between groups. Contextual SBMs (cSBMs) (Binkiewicz et al., 2017) extend SBM by relating community membership to node labels and allow control over homophily.

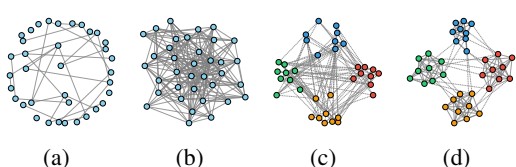

(a)    (b)    (c)    (d)

Figure 2: Network examples with various graph priors used in NodePFN training. (a) Sparse ER. (b) Dense ER. (c) Low homophily cSBM. (d) High homophily cSBM. For simplicity, class color-coded nodes are not shown in ER.

## 3 NODEPFN: PRIOR-FITTED NETWORKS FOR NODE CLASSIFICATION

We introduce NodePFN, a prior-fitted network that learns to approximate PPDs for node classification from synthetic graph data (see Fig. 3). Unlike traditional GNNs that require task-specific training, NodePFN performs in-context learning on arbitrary graphs in a single forward pass.

### 3.1 LEARNING FROM SYNTHETIC GRAPH PRIORS

Given the PPD framework from Section 2, we train a neural network $f_\theta$ to approximate posterior predictive distributions for node classification. During training, we sample synthetic graphs $\mathcal{G} \sim p(\mathcal{G})$ and learn to predict query node labels from context examples:

$$f_\theta : (\mathbf{x}_{\text{test}}, \mathcal{D}_{\text{train}}, \mathcal{G}) \mapsto p(\mathbf{y}_{\text{test}}|\mathcal{D}_{\text{train}}, \mathcal{G}), \tag{5}$$

where $\mathcal{D}_{\text{train}} = \{(\mathbf{x}_v, \mathbf{y}_v) : v \in \mathcal{V}_{\text{train}}\}$ contains labeled training nodes. This formulation naturally induces in-context learning: the model learns to extract patterns from training nodes and apply them to test nodes.

### 3.2 SYNTHETIC GRAPH PRIORS

As shown in Fig. 3(a), our approach begins with sampling diverse synthetic graph priors that capture the broad spectrum of structural patterns found in real-world networks.

**Feature-Label Relationships via Causal Models.** We generate feature-label relationships using structural causal models (SCMs) (Peters et al., 2017; Pearl, 2009) instantiated as random MLPs. For each graph, we sample an MLP architecture and convert it to a DAG by dropping random connections. Gaussian noise propagates through this network to produce node features $\mathbf{X}$ from intermediate layers and labels $y$ from later layers, creating complex non-linear dependencies. Importantly, for cSBM graphs, these generated labels determine the community assignments, which in turn control the graph structure through the homophily parameter $h$.

**Graph Structure Generation.** We use two random network models as shown in Fig. 2. (i) cSBMs generate graphs with controlled community structure and homophily. We sample the homophily level from 0.1 to 0.9. The cSBM creates edges with intra-community probability $p_{\text{in}}$ and inter-community

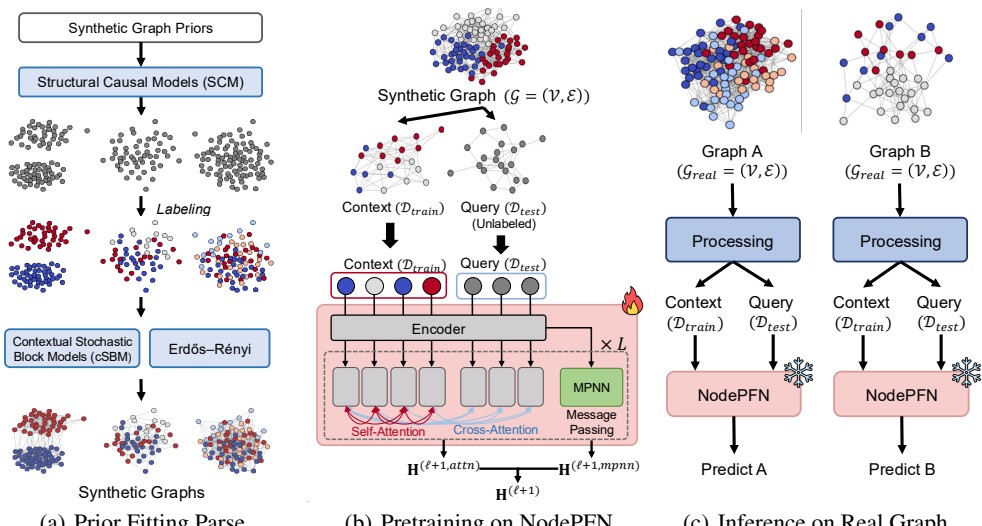

Figure 3: NodePFN overview. (a) Generation of diverse synthetic graph priors with varying structural properties. (b) Dual-branch architecture combining local message passing with context-query attention for in-context learning. (c) Inference on real-world graphs via the pre-trained NodePFN without task-specific re-training.

probability $p_{\text{out}}$ such that $h = p_{\text{in}}/(p_{\text{in}} + p_{\text{out}})$. This control over homophily allows us to generate graphs ranging from strong homophily to heterophily. (ii) ER networks provide unstructured baseline graphs where edges appear independently with probability $p_{\text{er}}$. This ensures the model learns beyond community-based patterns. The distribution for $p_{\text{er}}$ generates graphs with varying densities, from sparse to dense networks. During training, we sample from both networks to ensure comprehensive coverage of graph structures encountered in practice.

## 3.3 MODEL ARCHITECTURE

Each NodePFN layer consists of two parallel branches that process information complementarily, as shown in Fig. 3(b).

**Context-Query Attention Branch.** Following the PFN design (Müller et al., 2022), we use asymmetric attention patterns to enable in-context learning. The initial representations $\mathbf{H}_{\text{train}}^{(0)}$ combine embeddings of both features and labels, while $\mathbf{H}_{\text{test}}^{(0)}$ uses only feature embeddings (detailed implementation in Appendix B.5). For $\mathcal{V}_{\text{train}}$ with observed labels, self-attention allows them to build a comprehensive understanding of the label distribution:

$$\mathbf{H}_{\text{train}}^{(\ell+1,\text{attn})} = \text{SelfAttention}(\mathbf{H}_{\text{train}}^{(\ell)}, \mathbf{H}_{\text{train}}^{(\ell)}, \mathbf{H}_{\text{train}}^{(\ell)}). \quad (6)$$

For test nodes $\mathcal{V}_{\text{test}}$, cross-attention to training nodes enables leveraging the learned patterns:

$$\mathbf{H}_{\text{test}}^{(\ell+1,\text{attn})} = \text{CrossAttention}(\mathbf{H}_{\text{test}}^{(\ell)}, \mathbf{H}_{\text{train}}^{(\ell)}, \mathbf{H}_{\text{train}}^{(\ell)}), \quad (7)$$

where the attention functions follow the standard formulation. We employ multiple attention heads with outputs concatenated and linearly projected. This asymmetry ensures test nodes leverage training information without influencing each other's predictions.

**Local MPNN Branch.** In parallel, message passing aggregates neighborhood information to capture local graph topology:

$$\mathbf{H}^{(\ell+1,\text{mpnn})} = \text{MPNN}(\mathbf{H}^{(\ell)}, \tilde{\mathbf{A}}), \quad (8)$$

where $\tilde{\mathbf{A}} = \mathbf{D}^{-1/2}\mathbf{A}\mathbf{D}^{-1/2}$ is the symmetrically normalized adjacency matrix and $\mathbf{D}$ is a degree matrix. This branch captures structural patterns critical for classification regardless of train/test splits. In our framework, we use GCN (Kipf & Welling, 2017) for the local MPNN branch.

**Layer Fusion.** The parallel branches merge with the input via residual connections:

$$\mathbf{H}^{(\ell+1)} = \text{LayerNorm}(\mathbf{H}^{(\ell)} + \mathbf{H}^{(\ell+1,\text{attn})} + \mathbf{H}^{(\ell+1,\text{mpnn})}). \tag{9}$$

This design enables NodePFN to simultaneously learn from labeled examples via attention and local graph structure via message passing.

### 3.4 How to Train

We train NodePFN to approximate the PPD by minimizing the expected cross-entropy over synthetic graphs sampled from our prior:

$$\mathcal{L}(\theta) = \mathbb{E}_{\mathcal{D}\sim p(\mathcal{D})}\left[ -\frac{1}{|\mathcal{V}_{\text{test}}|} \sum_{v\in\mathcal{V}_{\text{test}}} \sum_{c=1}^{C} y_{v,c} \log f_\theta(y_{v,c}|\mathbf{x}_v, \mathcal{D}_{\text{train}}, \mathcal{G}) \right], \tag{10}$$

where $C$ is the number of classes, $y_{v,c}$ is the one-hot encoded label for node $v$ and class $c$, and $f_\theta$ is our neural approximation to the true PPD from Eq. (1). For each synthetic graph $\mathcal{G}$, we randomly partition nodes into $\mathcal{V}_{\text{train}}$ and $\mathcal{V}_{\text{test}}$.

### 3.5 How to Inference

As shown in Fig. 3(c), NodePFN performs direct prediction on a real-world graph $\mathcal{G}_{\text{real}}$ with its own training-test split. Given labeled nodes $\mathcal{V}_{\text{train}}$ with $\mathcal{D}_{\text{train}} = \{(\mathbf{x}_i, y_i) : i \in \mathcal{V}_{\text{train}}\}$ and unlabeled nodes $\mathcal{V}_{\text{test}}$, the model computes predictions in a single forward pass.

Given a real-world graph, we perform a preprocessing step (Appendix B.6) on the graph and its features. Then, the model processes the graph through $L$ NodePFN layers and outputs the PPD:

$$f_\theta(y_v|\mathbf{x}_v, \mathcal{D}_{\text{train}}, \mathcal{G}_{\text{real}}) = \text{softmax}(\mathbf{W}_{\text{out}}\mathbf{h}_v^{(L)}), \tag{11}$$

for each test node $v \in \mathcal{V}_{\text{test}}$. Training nodes incorporate label information through concatenation with features, while test nodes use only features. This provides calibrated uncertainty estimates as the model has learned to approximate the true PPD during training. Importantly, no gradient updates or fine-tuning are required — the pre-trained model generalizes directly to new graphs.

## 4 Experiments

In this section, we present experiments to evaluate the performance of our proposed NodePFN. We begin by detailing the experimental settings. Next, we investigate the following research questions:

- **(RQ1.)** Does our NodePFN perform well on various controlled homophily synthetic graphs?
- **(RQ2.)** Does our NodePFN generalize well for node classification on real-world benchmarks?
- **(RQ3.)** How does the performance of NodePFN compare against training-free methods?
- **(RQ4.)** Does NodePFN perform well compared to the baseline for structural node classification?
- **(RQ5.)** How do components contribute to NodePFN's effectiveness?

### 4.1 (RQ1.) Controlled Synthetic Graphs

**Setup.** To evaluate the classification capability on various homophily ratios, we use the synthetic Cora generator Li et al. (2021). The detailed synthetic datasets are in Appendix B.

**Results.** Fig. 4 shows the mean test accuracy. MLP maintains its test accuracy for all homophily rates. GCN and GAT perform poorly at low homophily rates. Our NodePFN has the best overall trend without sudden drops. Our prior data contribute to its stable accuracy for both homophily and heterophily settings compared with other models.

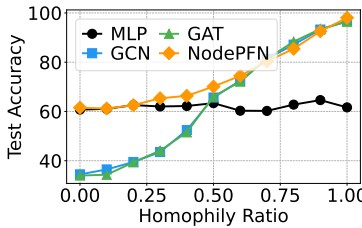

Figure 4: Experiments on the synthetic Cora Dataset.

Table 1: Performance comparison on homophily and heterophily real-world benchmark datasets. We report the average accuracy and ranking on each type of dataset, as well as the overall values.

| | Dataset | MLP | GCN | GAT | GraphAny (Products) | GraphAny (Arxiv) | GraphAny (Wisconsin) | GraphAny (Cora) | NodePFN |
|---|---|---|---|---|---|---|---|---|---|
| Homophily Graphs | AirBrazil | 23.08±5.83 | 42.31±7.98 | 57.69±14.75 | 34.61±16.54 | 34.61±16.09 | 36.15±16.68 | 33.07±16.68 | 75.38±1.88 |
| | AirEU | 21.25±2.31 | 41.88±3.60 | 32.50±8.45 | 41.75±6.84 | 41.50±6.50 | 41.13±6.02 | 40.50±7.01 | 57.00±1.21 |
| | AirUS | 22.88±1.46 | 46.49±1.81 | 48.47±4.17 | 43.57±2.07 | 43.64±1.83 | 43.86±1.44 | 43.46±1.40 | 61.66±0.31 |
| | Cora | 48.42±0.63 | 81.40±0.70 | 81.70±1.43 | 79.36±0.23 | 79.38±0.16 | 77.82±1.15 | 80.18±0.13 | 82.06±0.29 |
| | Citeseer | 44.40±0.44 | 63.40±0.63 | 69.10±1.59 | 67.94±0.29 | 68.34±0.23 | 67.50±0.44 | 68.90±0.07 | 67.30±0.83 |
| | Pubmed | 69.50±1.79 | 76.60±0.32 | 77.30±0.60 | 76.54±0.34 | 76.36±0.17 | 77.46±0.30 | 76.60±0.31 | 78.00±0.24 |
| | WikiCS | 72.72±0.43 | 79.12±0.45 | 79.27±0.20 | 75.01±0.54 | 74.95±0.61 | 73.77±0.83 | 74.39±0.71 | 75.98±0.80 |
| | Amazon-Photo | 68.20±0.88 | 91.88±0.79 | 91.86±1.07 | 90.64±0.82 | 90.60±0.82 | 90.18±0.91 | 90.14±0.93 | 90.53±0.13 |
| | Amazon-Comp | 58.28±2.98 | 85.83±0.86 | 87.01±0.50 | 82.90±1.25 | 83.04±1.24 | 82.00±1.14 | 82.99±1.22 | 81.42±0.48 |
| | DBLP | 56.27±0.62 | 73.02±2.22 | 73.87±1.35 | 70.62±0.97 | 70.90±0.88 | 70.13±0.77 | 71.73±0.94 | 74.71±0.39 |
| | Coauthor CS | 85.88±0.93 | 91.83±0.71 | 88.47±0.79 | 90.46±0.54 | 90.45±0.59 | 90.85±0.63 | 90.47±0.63 | 91.55±0.32 |
| | Coauthor Physics | 87.43±1.98 | 93.93±0.37 | 93.01±0.89 | 92.66±0.52 | 92.69±0.52 | 92.54±0.43 | 92.70±0.54 | 93.43±0.13 |
| | Deezer | 54.24±2.15 | 53.69±2.29 | 55.99±3.78 | 52.09±2.78 | 52.11±2.79 | 52.13±3.02 | 51.98±2.79 | 53.45±0.65 |
| | Average Accuracy | 56.43 | 73.05 | 74.39 | 71.09 | 71.14 | 70.86 | 71.45 | **77.39** |
| | Average Ranking | 7.62 | 4.92 | 4.54 | 4.46 | 4.31 | 4.31 | 4.15 | **1.69** |
| Heterophily Graphs | Cornell | 67.57±5.06 | 35.14±6.51 | 35.14±3.52 | 64.86±0.00 | 65.94±1.48 | 66.49±1.48 | 64.86±1.91 | 71.89±2.76 |
| | Texas | 48.65±4.01 | 51.35±2.71 | 54.05±2.41 | 73.52±2.96 | 72.97±2.71 | 73.51±1.21 | 71.89±1.48 | 76.22±7.53 |
| | Wisconsin | 66.67±3.51 | 37.25±1.64 | 52.94±3.10 | 65.89±2.23 | 65.10±3.22 | 71.77±5.98 | 61.18±5.08 | 79.22±6.97 |
| | Chameleon | 38.87±2.21 | 41.31±3.05 | 39.83±2.10 | 39.45±4.20 | 37.40±3.11 | 36.67±5.32 | 37.99±4.54 | 50.13±3.30 |
| | Actor | 33.95±0.80 | 28.55±0.68 | 27.30±0.22 | 28.99±0.61 | 28.60±0.21 | 29.51±0.55 | 27.91±0.16 | 32.99±1.09 |
| | Minesweeper | 80.00±0.00 | 81.12±0.37 | 80.08±0.04 | 80.27±0.16 | 80.30±0.13 | 80.13±0.09 | 80.46±0.15 | 80.66±0.25 |
| | Tolokers | 78.16±0.02 | 79.93±0.10 | 78.50±0.55 | 78.18±0.03 | 78.18±0.04 | 78.24±0.03 | 78.20±0.02 | 78.61±0.06 |
| | Amazon-Ratings | 47.90±0.45 | 47.35±0.26 | 47.18±0.42 | 42.70±0.10 | 42.74±0.12 | 42.57±0.34 | 42.84±0.04 | 44.68±0.48 |
| | Questions | 97.33±0.06 | 97.15±0.04 | 97.11±0.02 | 97.10±0.01 | 97.09±0.02 | 97.11±0.00 | 97.06±0.03 | 97.02±0.01 |
| | Squirrel | 35.55±0.98 | 38.67±1.84 | 38.78±2.39 | 38.92±2.98 | 37.73±2.31 | 36.76±3.55 | 37.25±2.65 | 43.40±1.03 |
| | Average Accuracy | 58.17 | 58.84 | 59.11 | 61.39 | 60.71 | 61.62 | 60.56 | **65.14** |
| | Average Ranking | 7.20 | 6.80 | 6.60 | 4.90 | 4.70 | 4.50 | 4.60 | **1.70** |
| | Avg. Accuracy | 57.30 | 66.63 | 67.67 | 66.24 | 65.93 | 66.24 | 66.00 | **71.27** |
| | Avg. Ranking | 7.41 | 5.86 | 5.57 | 4.68 | 4.50 | 4.40 | 4.38 | **1.70** |

## 4.2 (RQ2.) EXPERIMENTS ON REAL-WORLD GRAPH BENCHMARKS

**Setup.** We evaluate on 23 benchmark datasets for node classification. We compare against MLP, GCN (Kipf & Welling, 2017), GAT (Veličković et al., 2018), and GraphAny (Zhao et al., 2025) models. If there are reported results from Zhao et al. (2025), we directly adopt the reported results, otherwise, we run experiments with their optimal setting. More detailed dataset and evaluation settings are provided in Appendices A and B.7.

**Results.** Table 1 presents a comprehensive comparison of our results on 23 datasets. The results show that NodePFN achieves the best overall average accuracy of 71.27% using *only a single pre-trained model*. In contrast, GraphAny models require training on each specific dataset but still underperform NodePFN. On homophily and heterophily datasets, NodePFN achieves the highest average accuracy. Moreover, GraphAny models show inconsistent performance depending on the characteristics of the training dataset. GraphAny (Cora) performs well on homophily graphs but worse on Wisconsin, one of the heterogeneous graphs. In contrast, NodePFN consistently performs well on both graph types without requiring dataset-specific training.

## 4.3 (RQ3.) COMPARISON WITH TRAINING-FREE METHODS

**Setup.** NodePFN can be compared with several training-free methods, which can be tested directly without training steps. We compare closed-form solution methods that use pseudo-inverse operations to solve node classification as a regression problem (Zhao et al., 2025). The methods include the "Linear" model that predicts directly without graph

Table 2: Training-free models vs NodePFN.

| Method | Cora | Pubmed | Wisconsin | Texas |
|---|---|---|---|---|
| Linear | 52.80±0.00 | 59.30±0.00 | 80.00±2.15 | 32.35±5.30 |
| SGC | 78.20±0.00 | 72.98±0.00 | 57.64±1.07 | 46.03±6.86 |
| HGC | 22.50±0.00 | 46.32±0.00 | 64.32±2.51 | 57.54±6.30 |
| LabelProp | 60.30±0.00 | 63.44±0.04 | 16.08±2.15 | 23.53±5.51 |
| TFGNN | 60.03±0.00 | 40.04±0.01 | 14.51±3.01 | 19.91±6.10 |
| NodePFN | 82.06±0.29 | 78.00±0.24 | 81.18±5.70 | 76.22±7.53 |

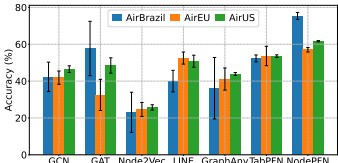

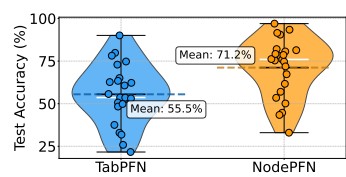

Table 3: Ablation studies on NodePFN components.

| Ablation | Cora | Wisconsin | Tolokers |
|---|---|---|---|
| w/o ER | 81.26 | 78.82 | 77.30 |
| w/o cSBM | 80.62 | 80.39 | 77.18 |
| TabPFN | 53.10 | 72.94 | 78.18 |
| NodePFN-L6 | 53.10 | 72.94 | 78.00 |
| NodePFN-Seq | 80.64 | 78.82 | 77.88 |
| NodePFN | 82.06 | 81.18 | 78.61 |

Figure 5: Structural node classification results (GraphAny trained on Wisconsin).

Figure 6: Comparison of accuracy distributions between TabPFN and NodePFN.

convolutions, and closed-form models using SGC (Wu et al., 2019) and high-pass filter graph convolutions (HGC) (Chien et al., 2021b; Luan et al., 2022). We also include label propagation ("LabelProp") (Zhu & Ghahramani, 2002) and TF-GNNs (Sato, 2024).

**Results.** In Table 2, the results demonstrate NodePFN's superior performance on all datasets compared to training-free baselines. Our NodePFN consistently outperforms all baseline methods while using a single pre-trained model. This demonstrates that the learned inductive bias of NodePFN surpasses analytical closed-form solutions and highlights the value of the pre-trained approach for generalizable node classification.

### 4.4 (RQ4.) STRUCTURAL NODE CLASSIFICATION

**Setup.** We evaluate NodePFN on Airport datasets (Ribeiro et al., 2017), where the goal is to predict the "structural role" of each node based only on the network topology without node features (Cui et al., 2022). Node features are provided as one-hot encoded node identifiers. This setting evaluates whether NodePFN can learn structural roles when forced to rely primarily on topological patterns. We include Node2Vec (Grover & Leskovec, 2016) and LINE (Tang et al., 2015) as additional baselines, since these methods specialize in structural embedding.

**Results.** As shown in Fig. 5, the results show that NodePFN outperforms all baselines. This suggests that NodePFN learns robust structural patterns that generalize beyond node features and effectively identifies meaningful node properties as well as structural roles based on network topology.

### 4.5 (RQ.5) ABLATION STUDIES

Table 3 demonstrates the robustness of NodePFN's design through ablation studies. Removing ER Networks or cSBM shows minimal performance degradation: for homophily datasets, Cora, cSBM removal causes slight drops, while for heterophily datasets such as Wisconsin and Tolokers (with 10 features and 0.5 homophily ratio), the impact is negligible. This indicates that these priors adapt well to different graph characteristics. The architectural variant NodePFN-L6, with reduced model capacity from 29.01M to 14.80M parameters, shows performance drops on Cora. This suggests that sufficient model capacity is important for learning patterns in highly homophily datasets. NodePFN-Seq with sequential processing maintains competitive performance, validating the effectiveness of both parallel and sequential architectures for combining structural information.

We also compare TabPFN, since when all graph-specific priors and MPNN components are removed, our proposed NodePFN can be reduced to TabPFN. As shown in Fig. 6, NodePFN outperforms TabPFN on all datasets (see Appendix D for full results). At the same time, TabPFN shows wider variance and lower overall accuracy, confirming the necessity of graph-aware modeling over treating nodes as independent tabular data.

## 5 RELATED WORK

**Prior-data Fitted Networks.** Müller et al. (2022) introduced PFNs and proved that a Transformer trained on tasks drawn from a prior can approximate PPDs from in-context examples. Following this work, Nagler (2023) shows how PFNs approximate PPD and why they can still learn at inference, and this paradigm has been adapted to specialized domains. TabPFN (Hollmann et al., 2023; 2025)

demonstrates that carefully designed synthetic priors can yield state-of-the-art performance on small tabular datasets. Also, the PFN has been adapted to time-series forecasting (Dooley et al., 2023) and Hoo et al. (2024; 2025) analyzes time series via feature engineering and encodes temporal patterns as tabular features. Concurrently, TabPFN-GN (Choi et al., 2025a) converts graph data into tabular features for direct use with TabPFN, but its reliance on feature engineering limits performance on heterophily graphs (cf. Section 4.5).

**Graph Foundation Models.** Recent works primarily leverage LLMs for zero-shot learning. GraphGPT (Tang et al., 2024), GraphLLM (Chai et al., 2023), and LLAGA (Chen et al., 2024a) convert graphs to text descriptions, while frameworks that use text-attributed graph datasets (Li et al., 2024c), such as OFA (Liu et al., 2024), use LLMs to encode node features. More recently, GOFA (Kong et al., 2024), Graph-R1 (Wu et al., 2025), and ZeroG (Li et al., 2024b) extend this line of work by exploring joint graph–language modeling, explicit reasoning for zero-shot learning, and cross-dataset transferability, respectively. These approaches leverage LLMs' strengths and limitations, including their dependency on textual attributes. In contrast, our approach requires no LLMs and works with arbitrary node features.

**GNNs for Node Classification.** Node classification is a classical graph machine learning task on which GNNs have recently achieved strong results. GCN (Kipf & Welling, 2017), Graph-SAGE (Hamilton et al., 2017), and GAT (Veličković et al., 2018) established the foundation of GNNs by showing strong performance on homophily graph datasets. Additionally, neighborhood aggregation of GNNs shows stable performance on homophily graph benchmark datasets but struggles with heterophily graphs (Pei et al., 2020). Sato (2024) proposes training-free GNNs for node classification, but they are always suboptimal for GNN performance and have limitations that make them inapplicable to heterophily graphs (see Section 4.3). As GNNs may not dominate all graph networks, zero-shot approaches leveraging pretrained models such as TabPFN can bypass this architecture search.

## 6 DISCUSSION

Although our primary objective is to demonstrate that universal node classification can be achieved via synthetic graph priors, the proposed NodePFN has limitations. NodePFN currently requires fixed maximum class numbers (tested up to 20 classes) and feature dimensions during training, and the attention mechanism's quadratic complexity restricts applicability to large-scale graphs. We leave these limitations for future work.

Despite these limitations, NodePFN's pre-training paradigm offers significant advantages. Although the model requires computational resources to pre-train on approximately 250,000 synthetic graphs (see Appendix C), this investment is amortized across all subsequent inference tasks. This contrasts with conventional GNNs that require retraining for each new dataset. Extensive experiments demonstrate that NodePFN achieves universal node classification, thereby justifying this initial computational overhead.

This universal applicability stems from our focus on learning structural patterns from synthetic priors. Unlike recent graph foundation models that rely on text-attributed graphs and LLMs (Tang et al., 2024; Chai et al., 2023; Liu et al., 2024) (as discussed in Section 5), NodePFN operates on graphs with arbitrary numerical features without requiring semantic understanding.

## 7 CONCLUDING REMARKS

We presented NodePFN, the first extension of PFNs to graphs, showing that universal node classification can be learned from synthetic graph priors. A single NodePFN model demonstrates an average accuracy of 71.27% on 23 benchmarks, particularly outperforming standard GNNs on heterophily graphs. This work represents a new paradigm for graph machine learning through synthetic pre-training, validating that universal patterns can be learned without real-world training data.

We discussed the limitations of NodePFN in Section 6. Future work could address these limitations by exploring efficient attention mechanisms for massive graphs and investigating hybrid approaches that combine our structural pattern learning with semantic processing from text-attributed graphs.

## ACKNWOLEDGEMENTS

This work was partly supported by the Institute for Information & Communications Technology Planning & Evaluation (IITP) grants funded by the Korean government (MSIT) (No. RS-2024-00457882, AI Research Hub Project; No. RS-2025-25442149, LG AI STAR Talent Development Program for Leading Large-Scale Generative AI Models in the Physical AI Domain), and the Korea Advanced Institute of Science and Technology (KAIST) grant funded by the Korea government (MSIT) (No. G04240001, Physics-inspired Deep Learning).

## ETHICAL STATEMENTS

In terms of the broader impact of this research on society, we do not see the very negative impacts that might be expected.

## USE OF LLMS

In accordance with ICLR 2026 policy, we acknowledge the use of LLMs in the preparation of this paper. To achieve perfect grammar and better expression and translation, we use Google Translate (Google LLC, 2025) and DeepL (DeepL SE, 2025) to improve some text. DeepL has an LLM-powered feature built in.

## REPRODUCIBILITY STATEMENT

To ensure reproducibility and completeness, we have included appendices in this paper. We also report the model architecture, all training hyperparameters, and the hardware specifications for our experiments in Appendix B. The synthetic graph prior generation and all associated hyperparameters are described in Appendices B.3 and B.7. The source code can be found in the following: `https://github.com/jeongwhanchoi/NodePFN`

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

# Supplementary Materials for "Learning Posterior Predictive Distributions for Node Classification from Synthetic Graph Priors"

## A  DETAILS OF DATASETS

**The Synthetic Cora Network**   The synthetic Cora dataset is provided by (Zhu et al., 2020b). Zhu et al. (2020b) generate graphs for a target homophily level using a modified preferential attachment process. We sample nodes, edges, and features from Cora to create a synthetic graph with a desired homophily and feature/label distribution. In Table 4, we summarize the properties of the synthetic Cora networks we used.

Table 4: The detailed information of the synthetic Cora. All levels of homophily have the same number of features (1,433), nodes (1,480), edges (5,936), and classes (5).

| Homophily | Avg. Degree | Max. Degree | Min. Degree |
|---|---|---|---|
| 0.0 | 3.98 | 84.33 | 1.67 |
| 0.1 | 3.98 | 71.33 | 2.00 |
| 0.2 | 3.98 | 73.33 | 1.67 |
| 0.3 | 3.98 | 70.00 | 2.00 |
| 0.4 | 3.98 | 77.67 | 2.00 |
| 0.5 | 3.98 | 76.33 | 2.00 |
| 0.6 | 3.98 | 76.00 | 1.67 |
| 0.7 | 3.98 | 67.67 | 2.00 |
| 0.8 | 3.98 | 58.00 | 1.67 |
| 0.9 | 3.98 | 58.00 | 1.67 |
| 1.0 | 3.98 | 51.00 | 2.00 |

**Real-world Graph Datasets.**   We list the dataset statistics we used in Tables 5 and 6. We use 23 benchmark datasets for node classification. Following prior work, these include both 13 homophily graphs (Kipf & Welling, 2017; Rozemberczki et al., 2021) (e.g., Cora, Citeseer, Pubmed, WikiCS) and 10 heterophilous graphs (Pei et al., 2020; Platonov et al., 2023b) (e.g., Cornell, Texas, Squirrel, Actor). For Chameleon and Squirrel, we use filtered datasets from Platonov et al. (2023a). We also report the clustering coefficients of each graph dataset.

Table 5: Homophily dataset statistics for node classification 13 benchmarks.

| Dataset | #Nodes | #Edges | #Features | #Classes | #Labels | Coeff. | Train/Val/Test (%) |
|---|---|---|---|---|---|---|---|
| AirBrazil | 131 | 1,074 | N/A | 4 | 80 | 0.6364 | 61.1/19.1/19.8 |
| AirEU | 1,190 | 5,995 | N/A | 4 | 80 | 0.5393 | 20.1/39.8/40.1 |
| AirUS | 10,008 | 13,599 | N/A | 4 | 80 | 0.5011 | 6.7/46.6/46.6 |
| Cora | 2,708 | 10,556 | 1,433 | 7 | 140 | 0.2407 | 5.2/18.5/36.9 |
| Citeseer | 3,327 | 9,104 | 3,703 | 6 | 120 | 0.1435 | 3.6/15.0/30.1 |
| Pubmed | 19,717 | 88,648 | 500 | 3 | 60 | 0.0602 | 0.3/2.5/5.1 |
| WikiCS | 11,701 | 431,206 | 300 | 10 | 580 | 0.4660 | 5.0/15.1/49.9 |
| Amazon-Photo | 7,650 | 238,162 | 745 | 8 | 160 | 0.4101 | 2.1/49.0/49.0 |
| Amazon-Comp | 13,752 | 491,722 | 767 | 10 | 200 | 0.3513 | 1.5/49.3/49.3 |
| DBLP | 17,716 | 105,734 | 1,639 | 4 | 80 | 0.1344 | 0.5/49.8/49.8 |
| Coauthor-CS | 18,333 | 163,788 | 6,805 | 15 | 300 | 0.3425 | 1.6/49.2/49.2 |
| Coauthor-Physics | 34,493 | 495,924 | 8,415 | 5 | 100 | 0.3776 | 0.3/49.9/49.9 |
| Deezer | 28,281 | 185,504 | 128 | 2 | 40 | 0.1412 | 0.1/49.9/49.9 |

Table 6: Heterophily dataset statistics for 10 benchmarks.

| Dataset | #Nodes | #Edges | #Features | #Classes | #Labels | Coeff. | Train/Val/Test (%) |
|---|---|---|---|---|---|---|---|
| Cornell | 183 | 554 | 1703 | 5 | 87 | 0.1671 | 47.5/32.2/20.2 |
| Texas | 183 | 558 | 1703 | 5 | 87 | 0.1979 | 47.5/31.7/20.2 |
| Wisconsin | 251 | 900 | 1703 | 5 | 120 | 0.2077 | 47.8/31.9/20.3 |
| Chameleon | 890 | 8854 | 2325 | 5 | 445 | 0.5769 | 50.0/25.0/25.0 |
| Actor | 7600 | 30,019 | 932 | 5 | 3648 | 0.0802 | 48.0/32.0/20.0 |
| Minesweeper | 10,000 | 78,804 | 7 | 2 | 5000 | 0.4355 | 50.0/25.0/25.0 |
| Tolokers | 11,758 | 1,038,000 | 10 | 2 | 5879 | 0.5329 | 50.0/25.0/25.0 |
| Amazon-Ratings | 24,492 | 186,100 | 300 | 5 | 12,246 | 0.5816 | 50.0/25.0/25.0 |
| Questions | 48,921 | 307,080 | 301 | 2 | 24,460 | 0.0307 | 50.0/25.0/25.0 |
| Squirrel | 2223 | 46,998 | 2089 | 5 | 1053 | 0.4631 | 50.0/25.0/25.0 |

# B    DETAILED EXPERIMENTAL SETTINGS

## B.1    HARDWARE AND SOFTWARE SPECIFICATIONS

Our implementation is developed on top of the TABPFN-V1[2] framework. All experiments were performed using the following software and hardware environments: UBUNTU 21.04 LTS, PYTHON 3.10.16, PYTORCH 1.12.1, PYTORCH GEOMETRIC 2.3.1, TORCH-SCATTER 2.1.2, TORCH-SPARSE 0.6.18, NUMPY 1.26.4, NETWORKX 3.3, SCIKIT-LEARN 1.4.0, CUDA 12.3, NVIDIA Driver 550.54.14, i9 CPU, NVIDIA RTX 6000.

## B.2    TRAINING SETUP

The model configuration of NodePFN is summarized in Table 7. In total, the model contains approximately 29.1M trainable parameters. We trained NodePFN for a total of 30 epochs, each epoch comprising up to 1,024 steps (245,760 steps in total) with a batch size of 8 (see more hyperparameters in Table 8. The total training required approximately 6 GPU hours on a single NVIDIA RTX A6000.

Table 7: Model configuration of NodePFN.

| Hyperparameter | Value |
|---|---|
| Embedding dimension | 512 |
| Number of layers | 12 |
| Number of attention heads | 4 |
| Dropout rate | 0.0 |

Table 8: Training hyperparameters for NodePFN.

| Hyperparameter | Value |
|---|---|
| Epochs | 30 |
| Steps per epoch | 1024 |
| Batch size | 8 |
| Embedding size | 512 |
| Learning rate | $\{1.5 \times 10^{-5},\ 5 \times 10^{-4},\ 1 \times 10^{-4}\}$ |
| Optimizer | AdamW |

## B.3    DETAILS OF NODEPFN PRIOR

**Structural Causal Models (SCM)**    We adopt the optimal sampling distributions from TabPFN (Holl-mann et al., 2023) for our SCM prior to ensure robust feature-label relationships. Following TabPFN's framework[3], each SCM is generated by:

---

[2]github.com/PriorLabs/TabPFN/tree/tabpfn_v1/
[3]https://github.com/PriorLabs/TabPFN/tree/tabpfn_v1

- Sampling MLP layers $\ell_{\text{scm}} \sim p(\ell_{\text{scm}})$ and hidden size $h_{\text{scm}} \sim p(h_{\text{scm}})$ from discretized noisy log-normal distributions

- Creating a layered graph structure and randomly dropping edges to form a DAG

- Selecting feature nodes and one label node from the causal graph

- Using activation functions sampled from Tanh, LeakyReLU, ELU, Identity

- Applying noisy log-normal noise distributions with beta-distributed dropout rates

This generates complex non-linear feature-label dependencies while maintaining the causal structure that has proven effective for tabular data modeling.

**Contextual SBM.**    Our contextual SBM generates community-structured graphs with controllable homophily levels:

- Sample homophily rate $h \sim \mathcal{U}(0.1, 0.9)$ and intra-community probability $p_{\text{in}} \sim \mathcal{U}(0.01, 0.1)$

- Compute inter-community probability as $p_{\text{out}} = p_{\text{in}} \times (1 - h)$

- Assign nodes to communities based on their labels from the SCM

- Generate edge probabilities using power distributions: $\text{probs}_{i,j} = \text{Power}(5) \times p_{\text{out}}$ for $i \neq j$ for inter-community edges, with diagonal values $p_{\text{out}} + \text{power}(2, size) \times (p_{\text{in}} - p_{\text{out}})$ for intra-community connections

- Create a symmetric probability matrix and generate edges via the stochastic block model

This approach ensures that network topology and node labels are inherently related through the homophily parameter.

**ER Network.**    ER networks provide unstructured baseline graphs complementing the community-based patterns:

- Sample edge probability $p_{\text{er}} \sim \mathcal{U}(0.01, 0.05)$

- Generate edges independently with probability $\mathcal{E}_{ij} \sim \text{Bernoulli}(p_{\text{er}})$

- Creates graphs with binomial degree distributions and no inherent community structure

The combination of cSBMs (50% of training graphs) and ER networks (50% of training graphs) ensures comprehensive coverage of graph structures from community-based patterns to random connectivity.

### B.4 Flexible Encoder for Variable Node Feature Dimensions

Our NodePFN model is designed to handle graphs with varying node feature dimensionalities up to a pre-defined maximum capacity. This is achieved through a flexible input encoder that standardizes the feature vectors before they are processed by the main architecture.

When a graph is provided where the node features have a dimension $d$ that is less than the maximum supported dimension of the model, $d_{max}$, each node's feature vector is first extended to $d_{max}$ dimensions by appending zero-padding. Then, to ensure that the zero-padding process does not systematically alter the input's scale or introduce bias, we apply a normalization factor to the padded vector. This mechanism ensures that feature vectors from different graph datasets are processed on a consistent scale, enabling our single pre-trained model to generalize effectively across a wide range of graph-structured data.

### B.5 Feature and Label Embeddings in Implementation

We employ learnable linear transformations $\mathbf{W}_{\mathcal{X}} \in \mathbb{R}^{d \times d_{\text{feat}}}$ and $\mathbf{W}_{\mathcal{Y}} \in \mathbb{R}^{d \times C}$ to project features and labels to the embedding dimension $d$, where $d_{\text{feat}}$ is the feature dimension and $C$ is the number

of classes. Following TabPFN-v1's implementation, we use element-wise addition rather than concatenation to combine features and labels:

$$\mathbf{H}^{(0)}_{\text{train}} = \mathbf{X}_{\text{train}}\mathbf{W}^{\top}_{\mathcal{X}} + \mathbf{Y}_{\text{train}}\mathbf{W}^{\top}_{\mathcal{Y}}, \quad \mathbf{H}^{(0)}_{\text{test}} = \mathbf{X}_{\text{test}}\mathbf{W}^{\top}_{\mathcal{X}},$$

where $\mathbf{X}_{\text{train}}$ and $\mathbf{Y}_{\text{train}}$ are the feature and label matrices for training nodes. This additive approach[4] maintains constant dimensionality and enables the model to learn complementary representations in different subspaces of the embedding vector. During inference, real-world labels are first converted to canonical integers before applying $\mathbf{W}_{\mathcal{Y}}$.

## B.6 INFERENCE IMPLEMENTATION DETAILS

We describe the data preprocessing methods used in the inference stage below.

**Graph Structure Preprocessing.** We convert the adjacency matrix of a given graph into a normalized adjacency matrix.

**Feature Preprocessing.** Adopting the ensemble approach from Hollmann et al. (2023), we employ a method that alters the order and scaling of features within the input context. This integrates a form of ensemble technique, using a fixed number of 32 multiple inputs, with the subsequent prediction results being aggregated. When real-world graphs have features exceeding the capacity of our NodePFN, we apply truncated SVD for dimensionality reduction. We also optionally apply feature smoothing using sum aggregation by edge connectivity for enhanced feature quality.

## B.7 EXPERIMENTAL SETTINGS FOR NODE CLASSIFICATION

**Evaluation Protocol.** For homophily datasets, we follow the semi-supervised setting of Kipf & Welling (2017) for Cora, Citeseer, and Pubmed (20 nodes per class for training, 500 validation, 1000 test), while for WikiCS, we follow the splits in Rozemberczki et al. (2021), and the remaining datasets follow the GraphAny (Zhao et al., 2025) protocol (20 nodes per class for training, the rest split evenly into validation and test). For heterophily datasets, we use the predefined split masks provided in Pei et al. (2020) and Platonov et al. (2023b). For Chameleon and Squirrel, we use filtered datasets from Platonov et al. (2023a).

**Search Space of Hyperparameters.** We report our search space of hyperparameters used in our experiments in Table 9. Note that we use the default hyperparameter of 32 ensembles.

Table 9: Homophily dataset hyperparameters for node classification.

| Hyperparam. | Range |
|---|---|
| # Components | {10,15,20,25,30} |
| # Smoothing Steps | {0, 1, 2, 3, 4} |

**Optimal Hyperparameters.** We report our optimal hyperparameters used in our experiments in Tables 10 and 11. We do not use truncated SVD on Tolokers and Minesweeper.

---

[4]https://github.com/PriorLabs/TabPFN/issues/93

Table 10: Homophily dataset hyperparameters for node classification.

| Dataset | # Components | # Smoothing Steps |
| --- | --- | --- |
| AirBrazil | 25 | 3 |
| AirEU | 25 | 1 |
| AirUS | 25 | 3 |
| Cora | 15 | 4 |
| Citeseer | 15 | 2 |
| Pubmed | 15 | 2 |
| WikiCS | 15 | 2 |
| Amazon-Photo | 15 | 3 |
| Amazon-Comp | 15 | 3 |
| DBLP | 25 | 2 |
| Coauthor-CS | 25 | 2 |
| Coauthor-Physics | 15 | 0 |
| Deezer | 25 | 2 |

Table 11: Heterophily dataset hyperparameters for node classification.

| Dataset | # Components | # Smoothing steps |
| --- | --- | --- |
| Cornell | 15 | 0 |
| Texas | 20 | 0 |
| Wisconsin | 25 | 0 |
| Chameleon | 25 | 0 |
| Actor | 35 | 0 |
| Minesweeper | - | 1 |
| Tolokers | - | 2 |
| Amazon-Ratings | 25 | 3 |
| Questions | 25 | 3 |
| Squirrel | 15 | 1 |

## C  PRIOR DATA SCALE ANALYSIS

We analyze the impact of the number of synthetic prior graphs on NodePFN performance and address the question of data efficiency in PFN training. This analysis aims to understand the trade-off between computational cost and performance gains when scaling prior data.

In our pretraining, each training iteration generates new synthetic graphs based on our random network prior, which creates diverse structural patterns. Fig. 7 reveals several key insights: (1) Particularly evident in the heterophily Texas dataset, its performance increases substantially with more prior data. The accuracy improves from approximately 53% to 76%. (2) For Cora, NodePFN shows more modest gains, and this suggests that patterns with higher homophily rates are effectively learned via cSBMs. (3) The performance improvement peaks at approximately 250,000 prior graphs.

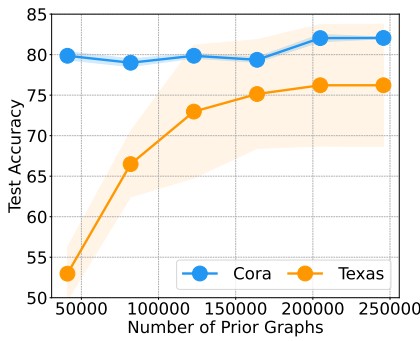

Figure 7: Impact of Prior Data Scale on NodePFN Performance

While generating synthetic prior data incurs an initial computational cost, this expense is amortized across all future inference tasks. Unlike traditional GNNs, which require retraining for each new graph, NodePFN enables immediate inference on arbitrary real-world graphs via one-time training on synthetic priors. Given the universal applicability of the resulting model, the computational investment in comprehensive prior generation proves worth it.

## D  COMPARISON WITH TABPFN AND NODEPFN

In Appendix D and Table 13, we report all results from Fig. 6 in Section 4.5. We used the TabPFN-v1 framework for comparison with TabPFN, and to ensure a fair comparison, we employed a pre-trained model using the same number of prior data points as ours. For TabPFN, we use the default hyperparameter of 32 ensembles, and we also enable feature subsampling for feature preprocessing.

Table 12: Comparison between TabPFN and NodePFN on homophily datasets (accuracy %).

| Dataset | TabPFN | NodePFN |
|---|---|---|
| AirBrazil | 52.31 | 75.38 |
| AirEU | 53.62 | 57.00 |
| AirUS | 53.55 | 61.66 |
| Cora | 53.10 | 82.06 |
| Citeseer | 32.94 | 67.30 |
| Pubmed | 65.04 | 78.00 |
| WikiCS | 62.25 | 75.98 |
| Amazon-Photo | 74.77 | 90.53 |
| Amazon-Comp | 50.66 | 81.42 |
| DBLP | 60.69 | 74.71 |
| Co-author CS | 48.32 | 91.55 |
| Co-author Physics | 63.41 | 93.43 |
| Deezer | 49.83 | 53.45 |
| **Average** | 55.62 | 77.39 |

Table 13: Comparison between TabPFN and NodePFN on heterophily datasets (accuracy %).

| Dataset | TabPFN | NodePFN |
|---|---|---|
| Cornell | 55.68 | 71.89 |
| Texas | 62.70 | 76.22 |
| Wisconsin | 72.94 | 79.22 |
| Chameleon | 37.54 | 50.13 |
| Actor | 25.84 | 32.99 |
| Minesweeper | 79.86 | 80.66 |
| Tolokers | 78.18 | 78.61 |
| Amazon-Ratings | 21.64 | 44.68 |
| Questions | 90.09 | 97.02 |
| Squirrel | 31.84 | 43.40 |
| **Average** | 53.47 | 65.14 |

# E    THEORETICAL DISCUSSION

The original PFN framework (Müller et al., 2022) establishes that Transformers can approximate posterior predictive distributions (PPDs) by minimizing the Prior-Data negative log-likelihood. Specifically, Müller et al. (2022, Insight 1) shows that this loss equals the expected cross-entropy between the true PPD and its approximation, while Müller et al. (2022, Corollary 1.2) guarantees convergence to the exact posterior under the infinite-capacity assumption, provided the architecture respects the exchangeability of the conditioning dataset $\mathcal{D}$. In practice, this requires the architecture to be permutation equivariant with respect to the ordering of training examples.

*Remark* E.1 (Preservation of PFN Guarantees in NodePFN). The dual-branch architecture of NodePFN maintains permutation equivariance because: the attention branch uses self-attention and cross-attention operations that are inherently permutation equivariant, the MPNN branch uses aggregation functions that are permutation equivariant, and their additive fusion (Eq. (9)) preserves this property. Therefore, NodePFN satisfies the exchangeability requirement of Müller et al. (2022, Corollary 1.2) and converges to the posterior.

This theoretical guarantee ensures that while the MPNN branch enriches the model with structural information, the fundamental Bayesian convergence properties of PFN remain intact.

# F    COMPARISON WITH HETEROPHILY-SPECIFIC GNNS

To further validate NodePFN's effectiveness on heterophily graphs, we compare against GNNs specifically designed for heterophily: H2GCN (Zhu et al., 2020b), GPRGNN (Chien et al., 2021a), and FAGCN (Bo et al., 2021). As shown in Table 14, NodePFN achieves the best performance on 7 out of 9 datasets despite using no real-world training data, while these specialized methods require dataset-specific training with carefully designed aggregation schemes for heterophily.

Notably, NodePFN shows improvements on Chameleon and Squirrel. The competitive performance on Texas and Actor within 1% of the best methods further confirms that learning from diverse synthetic graphs with controlled homophily provides generalization on the heterophily spectrum without requiring architectural modifications or dataset-specific tuning.

Table 14: Comparison of NodePFN with H2GCN, GPRGNN, FAGCN (accuracy %).

| Dataset | Chameleon | Squirrel | Cornell | Texas | Actor | Wisconsin | A.Ratings | Co.CS | Co.Physics |
|---|---|---|---|---|---|---|---|---|---|
| H2GCN | $41.07_{\pm 2.65}$ | $35.10_{\pm 1.15}$ | $65.77_{\pm 6.80}$ | $\mathbf{76.58_{\pm 1.56}}$ | $\mathbf{35.86_{\pm 1.03}}$ | $75.82_{\pm 1.13}$ | $40.87_{\pm 0.11}$ | $88.45_{\pm 0.97}$ | $92.86_{\pm 0.36}$ |
| GPRGNN | $39.69_{\pm 1.15}$ | $38.95_{\pm 1.99}$ | $40.54_{\pm 2.01}$ | $65.77_{\pm 1.56}$ | $33.94_{\pm 0.95}$ | $75.21_{\pm 4.08}$ | $42.23_{\pm 0.25}$ | $91.49_{\pm 0.39}$ | $92.76_{\pm 0.20}$ |
| FAGCN | $37.24_{\pm 3.54}$ | $36.78_{\pm 3.11}$ | $60.38_{\pm 1.82}$ | $68.44_{\pm 1.78}$ | $34.87_{\pm 1.25}$ | $72.02_{\pm 5.24}$ | $44.12_{\pm 0.31}$ | $91.07_{\pm 1.28}$ | $92.34_{\pm 0.40}$ |
| **NodePFN** | $\mathbf{50.13_{\pm 3.30}}$ | $\mathbf{43.40_{\pm 1.03}}$ | $\mathbf{71.89_{\pm 2.76}}$ | $76.22_{\pm 7.53}$ | $32.99_{\pm 1.09}$ | $\mathbf{79.22_{\pm 6.97}}$ | $\mathbf{44.68_{\pm 0.48}}$ | $\mathbf{91.55_{\pm 0.32}}$ | $\mathbf{93.43_{\pm 0.13}}$ |

# G    COMPARISON WITH LLM-BASED GRAPH METHODS ON GLBENCH

Following the experimental setting of recent LLM-based graph methods, we conduct the supervised node classification experiments on all the datasets in GLBench (Li et al., 2024c) [5].

NodePFN achieves competitive or superior performance compared to LLM-based graph foundation models without requiring text descriptions or language model dependencies. While LLM-based methods leverage pre-trained language knowledge, NodePFN leverages pre-trained patterns from massive synthetic prior data.

# H    COMPARISON WITH TABPFN USING SMOOTHED FEATURES

Tables 16 and 17 present comprehensive comparisons between TabPFN-v1 (Hollmann et al., 2023) with smoothed features and NodePFN across homophily and heterophily datasets. The smoothed

---

[5]https://github.com/NineAbyss/GLBench

Table 15: Accuracy under the supervised setting of GLBench (Li et al., 2024c). **Best** and second-best are highlighted.

| Dataset | Cora | Citeseer | Pubmed | WikiCS |
|---|---|---|---|---|
| InstructGLM (Ye et al., 2023) | 69.10 | 51.87 | 71.26 | 45.73 |
| GraphText (Zhao et al., 2023) | 76.21 | 59.43 | 75.11 | 67.35 |
| LLaGA (Chen et al., 2024a) | 74.42 | 55.73 | 68.82 | 73.88 |
| OFA (Liu et al., 2024) | 75.24 | **73.04** | **75.61** | 77.34 |
| **NodePFN** | **76.38** | 63.08 | 68.18 | **76.29** |

features baseline applies non-parametric feature aggregation before feeding node representations into TabPFN-v1's official checkpoint[6], effectively incorporating local neighborhood information without explicit graph structure modeling. On homophily datasets (Table 16), NodePFN demonstrates consistent improvements. The advantages become even more pronounced on heterophily datasets (Table 17), where NodePFN substantially outperforms the smoothed feature baseline on Cornell , Texas, and Wisconsin. These results validate that NodePFN's explicit modeling of graph topology through its dual-branch architecture provides meaningful improvements over simple feature smoothing approaches. Note that TabPFN-v1's limitation to 10 classes prevents evaluation on datasets like Co-author CS, whereas NodePFN supports up to 20 classes.

Table 16: Comparison between TabPFN with smoothed features and NodePFN on homophily datasets (accuracy %).

| Dataset | TabPFN-v1 (smoothed features) | NodePFN |
|---|---|---|
| AirBrazil | 67.69 | 75.38 |
| AirEU | 55.62 | 57.00 |
| AirUS | 59.60 | 61.66 |
| Cora | 74.06 | 82.06 |
| Citeseer | 51.16 | 67.30 |
| Pubmed | 75.96 | 78.00 |
| WikiCS | 74.90 | 75.98 |
| Amazon-Photo | 83.69 | 90.53 |
| Amazon-Comp | 75.61 | 81.42 |
| DBLP | 69.20 | 74.71 |
| Co-author CS | N/A | 91.55 |
| Co-author Physics | 87.93 | 93.43 |
| Deezer | 48.17 | 53.45 |

Table 17: Comparison between TabPFN-v1 with smoothed features and NodePFN on heterophily datasets (accuracy %).

| Dataset | TabPFN-v1 (smoothed features) | NodePFN |
|---|---|---|
| Cornell | 42.16 | 71.89 |
| Texas | 56.22 | 76.22 |
| Wisconsin | 51.37 | 79.22 |
| Chameleon | 41.42 | 50.13 |
| Actor | 25.29 | 32.99 |
| Minesweeper | 80.07 | 80.66 |
| Tolokers | 78.05 | 78.61 |
| Amazon-Ratings | 44.24 | 44.68 |
| Questions | 97.02 | 97.02 |
| Squirrel | 40.42 | 43.40 |

## I    EXTENSIVE ABLATIONS ON SYNTHETIC PRIOR DESIGN

**Graph generation models and homophily distribution.**    Table 18 shows ablation results on different graph generation models and homophily distributions. We compare our approach against various alternatives including using only ER graphs, only cSBM graphs with restricted or full homophily ranges, and only Barabási-Albert (BA) networks (Barabási & Albert, 1999). As shown in Table 18, the results reveal several key insights. First, restricting training to specific homophily ranges leads to performance degradation. This shows the necessity of covering the full homophily spectrum to generalize across diverse real-world graphs. Second, training exclusively on Barabási-Albert networks — which explicitly model power-law degree distributions — shows inconsistent performance. This suggests that power-law topology alone provides insufficient structural diversity for universal graph learning. Third, using only ER or only cSBM graphs underperforms the combined approach, validating that both graph types contribute complementary inductive biases.

---

[6] https://github.com/PriorLabs/TabPFN/tree/tabpfn_v1

Table 18: Ablation study on graph generation models and homophily distributions.

| Ablation | Cora | Wisconsin | Tolokers |
|---|---|---|---|
| Only ER | 80.62 | 80.39 | 77.18 |
| Only cSBM (0.1-0.3) | 79.89 | 79.98 | 77.65 |
| Only cSBM (0.7-0.9) | 80.42 | 78.20 | 77.23 |
| Only cSBM (full range) | 81.26 | 78.82 | 77.30 |
| Only BA (Barabási-Albert) | 74.18 | 80.57 | 74.63 |
| **NodePFN (ER+cSBM)** | **82.06** | **81.18** | **78.61** |

**ER/cSBM ratio analysis.** Table 19 examines how the mixture ratio between ER and cSBM graphs affects performance. We varied the proportion of ER graphs from 0% (cSBM only) to 100% (ER only). The balanced 50/50 ratio consistently achieves optimal or near-optimal performance across all homophily regimes. Notably, heterophilic Wisconsin benefits from higher ER ratios (80-50% range), likely because ER's unbiased topology provides crucial structural diversity for heterophilic learning, while homophilic Cora shows more robustness to varying ratios. These results show that ER's unbiased topology and cSBM's community structure provide complementary inductive biases essential for universal graph learning.

Table 19: Ablation study on ER/cSBM mixture ratio.

| ER Ratio | Cora | Wisconsin | Tolokers |
|---|---|---|---|
| 100% (Only ER) | 80.62 | 80.39 | 77.18 |
| 80% | 80.90 | 81.30 | 77.20 |
| **50% (NodePFN)** | **82.06** | **81.18** | **78.61** |
| 20% | 82.01 | 78.75 | 78.10 |
| 0% (Only cSBM) | 81.26 | 78.82 | 77.30 |

## J    ARCHITECTURAL ABLATIONS

We provide comprehensive ablation studies on NodePFN's dual-branch architecture to demonstrate the necessity and contribution of each component. Table 20 shows results for different architectural variants compared to the full NodePFN model.

Removing the MPNN branch causes substantial degradation on both homophilic Cora and heterophilic Wisconsin, demonstrating that the MPNN provides essential structural inductive biases that pure attention cannot capture. NodePFN-Seq underperforms the parallel design. Reducing model capacity to 6 layers (NodePFN-L6) causes failure on Cora, demonstrating that sufficient depth is important for learning diverse patterns from many synthetic priors. These ablations validate that the MPNN provides structural biases, and their parallel combination enables optimal integration, and a deep architecture is required.

Table 20: Ablation study on architectural design choices.

| Ablation | Cora | Wisconsin | Tolokers |
|---|---|---|---|
| NodePFN-L6 | 53.10 | 72.94 | 78.00 |
| NodePFN-Seq | 80.64 | 78.82 | 77.88 |
| NodePFN w/o MPNN | 75.50 | 70.10 | 78.09 |
| **NodePFN (Full)** | **82.06** | **81.18** | **78.61** |

## K    STATISTICS OF SYNTHETIC PRIOR DATASETS

We provide detailed statistics of the synthetic graphs used for pre-training NodePFN. Our training set consists of approximately 250,000 graphs generated over 30 epochs, with each graph sampled from a

mixture of ER networks and cSBM with varying homophily ratios. All synthetic graphs are fixed at 1,024 nodes to balance computational efficiency with sufficient structural complexity for learning meaningful patterns. The number of classes per graph varies from 1 to 20, and edge counts vary based on the underlying generation model and density parameters. On average, each graph contains 12,706.4 edges and 8.79 classes.

Fig. 8 shows the distributions of edges and classes across all synthetic graphs. The edge distribution (Fig. 8(a)) exhibits a normal distribution centered around 10,000-15,000 edges, with a long tail extending to sparser graphs. This diversity reflects the combination of sparse ER networks, which generate fewer edges on average, and dense cSBM communities, which create higher edge densities within communities. The class distribution (Fig. 8(b)) shows coverage across the full range of 1-20 classes, with slightly higher frequency for graphs with fewer classes.

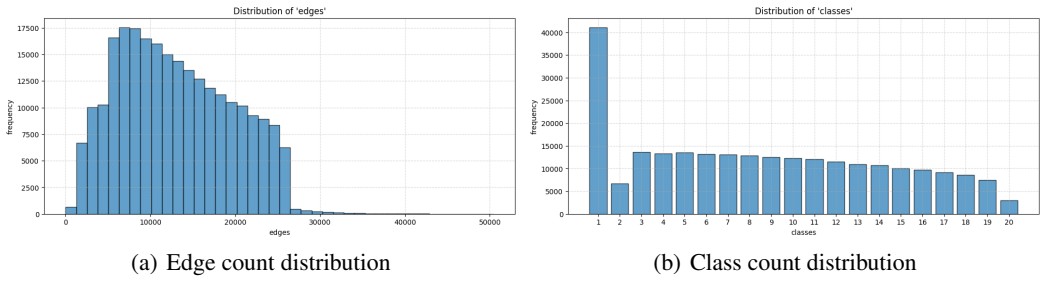

(a) Edge count distribution         (b) Class count distribution

Figure 8: Distribution of synthetic prior datasets used for pre-training.

## L    COMPUTATIONAL COMPLEXITY

We provide formal complexity analysis for NodePFN. For a graph with $N$ nodes, $|E|$ edges, $d$-dimensional features, and $L$ MPNN layers, the MPNN branch requires $\mathcal{O}(LEd)$ operations for message passing and aggregation, identical to standard GCN complexity. The Transformer branch computes attention over all nodes, requiring $\mathcal{O}(N^2d)$ operations for attention computation and $\mathcal{O}(Nd^2)$ for feed-forward layers. The total per-graph complexity is therefore $\mathcal{O}(LEd + N^2d)$.

Table 21 shows comprehensive runtime comparison between GCN and NodePFN across all 23 benchmark datasets. While NodePFN achieves superior average accuracy and ranking, it also demonstrates remarkable computational efficiency in terms of total deployment cost. GCN requires cumulative training time of 188 seconds plus 12.35 seconds for inference across all datasets. Importantly, this represents a single training run per dataset with fixed hyperparameters — in practice, achieving competitive performance typically requires multiple hyperparameter tuning attempts, potentially multiplying this cost by 5 to 10 times or more. In contrast, NodePFN requires only 47.78 seconds total representing a 4 times speedup in total time-to-deployment. The one-time pre-training cost (6 GPU hours) is amortized across unlimited datasets, eliminating repetitive per-dataset optimization and making it increasingly efficient as more graphs are processed.

Table 21: Runtime efficiency comparison

| 23 Benchmark Datasets | GCN | NodePFN |
|---|---|---|
| Avg. Accuracy | 66.63 | 71.27 |
| Avg. Ranking | 5.86 | 1.70 |
| Total Train Runtime ($s$) | 188 | 47.78 |
| Total Predict Runtime ($s$) | 12.35 | |

