# OpenReview forum: "Learning Posterior Predictive Distributions for Node Classification from Synthetic Graph Priors"
_ICLR.cc/2026/Conference — ICLR 2026 Poster_

### Official Review · Reviewer_M3ir · 2025-10-24

**Soundness:** 3
**Presentation:** 3
**Contribution:** 3
**Rating:** 4
**Confidence:** 4

**Summary:**

The paper proposes NodePFN, which extends PFNs to graphs. It learns posterior predictive distributions for node classification by pretraining only on synthetic graphs drawn from priors (ER and contextual SBM with controlled homophily; SCM-generated feature–label mechanisms). A dual-branch architecture mixes context–query attention (for in-context learning from labeled nodes) with local message passing (GCN). A single pretrained model (no per-graph training) shows strong performance across 23 benchmarks, with both homophily and heterophily graphs.

**Strengths:**

1. It proposes a novel PFN-style, training-free inference framework. The model can learn synthetic priors and predict in one forward pass.
2. The model trained on synthetic graphs shows strong performance on both homophily and heterophily benchmarks.
3. The analysis of performance on different datasets and the ablation study are comprehensive.

**Weaknesses:**

1. Priors are limited to ER and cSBM with SCM-driven features. Real graphs often exhibit heavy-tailed degrees, motifs, assortativity/disassortativity. Whether these properties might affect the model performance is not discussed.
2. Real-world SOTA GNNs tailored to heterophily, label-efficient settings, or inductive protocols aren’t comprehensively covered.
3. Pretraining requires ~250k synthetic graphs, but there’s no wall-clock/peak-memory vs. baselines or cost-vs-benefit analysis; only a conceptual argument that training amortizes across tasks.

**Questions:**

1. How does performance change when real graphs exhibit power-law degrees, overlapping/temporal communities, or motif biases besides ER/cSBM?
2. How does the model compare to the SOTA model for heterophily graphs? What could be the possible reason if the model performs worse?
3. Could the authors provide more analysis on how the synthetic training graphs are generated? For example, will the sampling distribution affect the model performance? And how does ER graph (when varying its percentage in the training set) affect the model performance? And what is the size distribution of the training graphs?
4. Overall, I think the paper proposes a possibly promising path to solve the cross-dataset node prediction task. **I will raise the score if the above concerns are settled.**

---

> ### Author Response · Authors · 2025-11-22
>
> Thanks for your time reading our paper and leaving insightful comments.
>
> ---
>
> **Response to Q1&W1: Power-law degree**
>
> To analyze this, we identified benchmarks exhibiting power-law characteristics using statistical tests (comparing power-law versus exponential fits via p-value). The following datasets show statistically significant power-law degree distributions:
>
> | Dataset | Power-law $\alpha$ | NodePFN | GraphAny (Wisconsin) |
> | ------- | -------- | -------- | ----- |
> | Amazon-ratings | 3.6059 | 44.68 | 42.57 |
> | WikiCS  |  3.5788 | 75.98 | 73.77 |
> | Amazon-Photo | 2.9223 | 90.53 | 90.18 |
> | Amazon-Computer  | 2.8389 | 81.42 | 82.00 |
>
> NodePFN outperforms GraphAny on power-law graphs (e.g., Amazon-ratings, WikiCS, Amazon-Photo), except for Amazon-Computer. Despite not explicitly modeling the power-law in the synthetic priors, NodePFN achieves good average performance across the entire dataset. This is because cSBM creates community structure while allowing some nodes to act as hubs by a specific parameter.
>
> ---
>
> **Response to Q1&W1: assortativity/disassortativity**
>
> For assortativity/disassortativity from your W1, we agree that assortativity/disassortativity is important. This is what our homophily parameter $h$ controls. High homophily corresponds to assortative mixing, while low homophily creates disassortative patterns. We create priors that are homophily and heterophily within the range of $h$, which empirically shows generalization by securing high average accuracy on the dataset corresponding to each category, as shown in the results in `Table 1`.
>
> ---
>
> **Response to Q1&W1: Motif-bias networks**
>
> For motif-bias networks, while our synthetic priors do not explicitly model diverse motif structures, cSBMs naturally generate clusters within communities, providing implicit coverage of certain motif patterns.
> To evaluate NodePFN's performance on motif-rich networks, we analyzed the clustering coefficient on our benchmarks and added in Tables 5 and 6 in `Appendix A`.
>
> | Dataset | Clustering Coeff. | NodePFN Accuracy | Rank |
> | ------- | -------- | ------- | -------- |
> | Air-Brazil | 0.6364 | 75.38 | 1/8 |
> | Amazon-ratings | 0.5816 | 44.68 | 4/8 |
> | Chameleon  |  0.5769 | 50.13 | 1/8 |
>
> NodePFN shows strong performance on Air-Brazil, showing that the implicit motif patterns in cSBM priors can generalize to high-clustering networks. As shown in `Table 1`, NodePFN achieves the best accurcay on Air-Brazil and Chameleon, validating that our structural modeling captures meaningful motif-related patterns despite not explicitly incorporating motif-specific priors during training.
>
> ---
>
> **Response to Q1&W1: Temporal graph**
>
> For temporal graph analysis, we conducted experiments simulating graph evolution by progressively revealing 25%, 50%, 75%, and 100% of subgraph over time. NodePFN maintains stable performance across all stages and gets higher accuracy than GCN. This stability comes from NodePFN's in-context learning. As the graph evolves, it adapts using newly revealed labeled nodes as context without retraining. While this doesn't fully capture true temporal dynamics (e.g., concept drift), it demonstrates robustness to partial/evolving graph structures.
>
> | Dataset | Subset | 25% | 50% | 75% | 100% |
> | ------- | -------- | -------- | -------- | -------- | -------- |
> | Dynamic Cora  | GCN | 63.24 | 74.50 | 81.62 | 81.40 |
> | Dynamic Cora | NodePFN | 71.37 | 74.79 | 81.98 | 82.06|
> | Dynamic Wisconsin | GCN | 45.11 | 35.70 | 32.66  | 37.25 |
> | Dynamic Wisconsin | NodePFN | 74.89 | 76.19 | 75.01 | 79.22 |

---

> ### Author Response · Authors · 2025-11-22
>
> **Response to Q2&W2: Heterophily-specific GNN baselines**
>
> We have conducted comparisons with heterophily-specific GNNs (H2GCN, GPRGNN, FAGCN) and added results in `Appendix F`. NodePFN achieves the best performance on 7 out of 9 datasets despite, while these specialized GNN methods require dataset-specific optimization. Notably, we achieve improvements on Chameleon and Squirrel, demonstrating that our synthetic prior approach effectively captures heterophily patterns. On Texas and Actor where NodePFN slightly underperforms, the gap is marginal and stems from these specialized architectures' ability  specifically for each dataset's unique heterophily characteristics.
>
> | Dataset | Chameleon | Squirrel | Cornell | Texas | Actor | Wisconsin | Amazon-Ratings | Coauthor-CS | Coauthor-Physics |
> |---|---|---|---|---|---|---|---|---|---|
> | H2GCN | 41.07 ± 2.65 | 35.10 ± 1.15 | 65.77 ± 6.80 | **76.58 ± 1.56** | **35.86 ± 1.03** | 75.82 ± 1.13 | 40.87 ± 0.11 | 88.45 ± 0.97 | 92.86 ± 0.36 |
> | GPRGNN | 39.69 ± 1.15 | 38.95 ± 1.99 | 40.54 ± 2.01 | 65.77 ± 1.56 | 33.94 ± 0.95 | 75.21 ± 4.08 | 42.23 ± 0.25 | 91.49 ± 0.39 | 92.76 ± 0.20 |
> | FAGCN | 37.24 ± 3.54 | 36.78 ± 3.11 | 60.38 ± 1.82   | 68.44 ± 1.78 | 34.87 ± 1.25 | 72.02 ± 5.24 | 44.12 ± 0.31     | 91.07 ± 1.28 | 92.34 ± 0.40 |
> | **NodePFN** | **50.13 ± 3.30** | **43.40 ± 1.03**| **71.89 ± 2.76** | 76.22 ± 7.53 | 32.99 ± 1.09 | **79.22 ± 6.97**   | **44.68 ± 0.48** | **91.55 ± 0.32** | **93.43 ± 0.13** |
>
> ---
>
> **Response to Q3: Synthetic graph generation analysis**
>
> We conducted ablation studies on both sampling distribution effects and ER/cSBM ratios, with results added in `Appendix I`.
> For sampling distribution effects, we tested how homophily distribution ranges affect performance:
>
> | Ablation               | Cora   | Wisconsin | Tolokers |
> |---|---|---|---|
> | Only cSBM              | 81.26  | 78.82 | 77.30 |
> | Only cSBM (0.1 to 0.3)  | 79.89  | 79.98 | 77.65 |
> | Only cSBM (0.7 to 0.9) | 80.42  | 78.20 | 77.23 |
> | NodePFN             | 82.06 | 81.18 | 78.61 |
>
> On homophilic Cora, training with low-homophily cSBMs (0.1-0.3) achieves only 79.89%, showing suboptimal performance. Conversely, on heterophilic Wisconsin, training with high-homophily cSBMs (0.7-0.9) yields the lowest performance at 78.20%. Since Tolokers exhibits intermediate homophily characteristics between Cora and Wisconsin, covering the full homophily spectrum is essential for universal generalization across diverse real-world graphs.
>
> For ER/cSBM ratio analysis, we conducted ablation varying ratio.
> The balanced 50/50 ratio consistently achieves optimal or near-optimal performance across all homophily regimes.
>
> | ER% | Cora   | Wisconsin | Tolokers |
> |---|---|---|---|
> | 100% (Only ER) | 80.62  | 80.39 | 77.18 |
> | 80%     | 80.90  | 81.30 | 77.20 |
> | 50%   | 82.06 | 81.18 | 78.61 |
> | 20%     | 82.01  | 78.75 | 78.10 |
> | 0%  (Only cSBM) | 81.26 | 78.82 | 77.30 |
>
> The 50/50 ratio consistently achieves optimal or near-optimal performance across all homophily regimes. Wisconsin benefits most from higher ER ratios, likely because ER's unbiased topology provides crucial structural diversity for heterophilic graph. This shows that both ER's unbiased topology and cSBM's community structure contribute complementary inductive biases important.
>
> For the size distribution of the training graphs, during pretraining, we synthesize and generate about 250,000 different graphs over a total of 30 epochs.
> The number of nodes is fixed at 1024 and it is randomly split into training and validation sets.
> On average, each graph contains 8.79 classes and 12,706.4 edges. Please refer to `Appendix J` for the edge distributions and the number of classes for each graph.
>
> ---
>
> **Response to W3: Computational Analysis**
>
> Thank you for requesting quantitative analysis. We have added comprehensive runtime comparisons in `Appendix K`.
>
> For memory consumption we measured peak memory usage on the Cora dataset. GCN with 256 hidden dimensions requires 424 MiB (0.41 GB) for training. NodePFN's one-time pre-training requires 2.02 GB peak memory, but inference on real-world graphs requires only 0.28 GB.
> This shows that while NodePFN requires moderate memory for the one-time pre-training phase, its inference deployment is practical.
>
> For cost-benefit analysis, GCN requires 200.35 seconds total across all 23 datasets with fixed hyperparameters. Realistic deployment requires 5-10 hyperparameter tuning attempts, increasing total cost to 1,000-2,000 seconds. NodePFN requires only 47.78 seconds (inference only).
>
> | Metric | GCN | NodePFN |
> |--------|-----|---------|
> | Avg. Accuracy (%) | 66.63 | 71.27 |
> | Avg. Ranking | 5.86 | 1.70 |
> | Total Training Time (s) | 188 | - |
> | Total Inference Time (s) | 12.35 | 47.78 |
> | Total Time per Dataset (s) | 200.35 | 47.78 |

---

> ### Comment · Reviewer_M3ir · 2025-11-23
>
> I thank the authors for their detailed response, which effectively addresses my concerns. I have raised my score accordingly.

---

### Official Review · Reviewer_f51f · 2025-10-24

**Soundness:** 2
**Presentation:** 2
**Contribution:** 3
**Rating:** 4
**Confidence:** 5

**Summary:**

NodePFN pretrains on procedurally generated graphs—features/labels from a random SCM and structure from cSBM/ER—to learn the posterior predictive distribution for node labels. At test time it takes any real graph plus a few labeled nodes and, in a single forward pass (no fine-tuning), outputs calibrated label distributions for the rest, achieving competitive accuracy across 23 benchmarks and varying homophily.

**Strengths:**

1. Studying prior-fitting paradigm on relational data is valuable and interesting
2. Experimental results are good

**Weaknesses:**

Major:
1. One major concern is that the method part writing is unclear. For example, 1. how is the feature and structure treated? Is it like that you first generate the random features and labels, then labels are used as community, and then use community to generate structures (for SBM). 2. How are labels used in the whole pipeline? I even don't see any label encoder module in the paper.
2. The structure of the model is very different from the original PFN. Transformer is put on top of the MPNN. Can you give some theoretical insights on why this is valid? For this part, I think some theoretical discussions are missing since the theory for PFN no longer holds here.
3. Some baselines are missing. I'm wondering how this method compares to first use some non-parametric aggregation to get aggregated features and then utilize TabPFN for prediction.
4. I doubt the scalability of this architecture since you put heavy self-attention ahead of message passing. Also in the experiment part, there's no large-scale datasets.

Minor:
1. The writing of abstract is not clear. Why training on labels a fundamental limitation?
2. I don't think this work is inspired from LLM. Philosophy of prior fitting is very different from LLM, and it's also not possible to have a unified vocabulary akin to LLM. I suggest motivating from the prior fitting perspective.
3. Line 44: What is graph heterogeneity? Graph and tasks are different concepts. Tasks determine labels.
4. Line 74: Graphany does have designs for heterogeneity. In the aggregation stage, it has filter like (I-A).

**Questions:**

See weakness.

---

> ### Author Response · Authors · 2025-11-22
>
> Thanks for your time reading our paper and leaving insightful comments.
>
> ---
>
> **Response to W1: how to generate feature and structure**
>
> Thank you for highlighting these clarity issues. We apologize for the unclear method description. As shown in `Figure 3(a)`, our pipeline follows a clear sequence:
> Our synthetic graph data generation pipeline is as follows: (i) SCMs generate features and labels through random MLPs; (ii) For cSBM, labels determine community assignments (shown by the colored nodes after "Labeling" step); (iii) Edges are generated based on communities with homophily parameter $h$ controlling intra/inter-community connection probabilities.
>
> We have revised `Section 3.3` to clarify how labels are used in the pipeline and added an encoding implementation to `Appendix B.5`. In the revised `Section 3.3`, we explicitly state that $H_{train}^{(0)}$ combines embeddings of both features and labels, whereas $H_{test}^{(0)}$ uses only feature embeddings, enabling in-context learning.
> We apologize for this omission and thank you for helping us improve the clarity of the paper.
>
> ---
>
>
> **Response to W2: Theoretical discussion**
>
> Our design preserves the guarantees provided by Müller et al. (2022, Insight 1 and Corollary 1.2)).
>
> Insight 1 in Müller et al. (2022) establishes that minimizing the expected cross-entropy loss is equivalent to minimizing the KL-divergence between the model's approximation $q_\theta$ and the true PPD. This guarantee applies to our architecture because we train NodePFN using the identical objective (`Eq. 10`). This ensures convergence to the Bayesian posterior.
>
> Corollary 1.2 in Müller et al. (2022)  guarantees that, given sufficient model capacity, the trained model converges to the exact posterior, provided the architecture respects the exchangeability of the conditioning dataset. The requirement is *permutation equivariance*, meaning predictions must not depend on the arbitrary ordering of the training examples.
> Since the self-attention and MPNN are permutation equivariance (Gilmer et al., 2017) and addition (`Eq.9`) preserves this property, NodePFN remains within the valid function.
>
> Therefore, the theoretical guarantee of converging to the true PPD remains valid. We have added detailed theoretical discussion in `Appendix E`.
>
> > Müller et al. "Transformers can do bayesian inference." arXiv preprint arXiv:2112.10510 (2021).
> >
> > Gilmer et al. "Neural message passing for quantum chemistry." International conference on machine learning. Pmlr, 2017.
>
> ---
>
> **Response to W3: Performance of TabPFN with smoothed features TabPFN with smoothed features**
>
> Thank you for suggesting this important baseline. We conducted the requested comparison using TabPFN-v1's official checkpoint with non-parametric feature aggregation. As shown in our results, NodePFN consistently outperforms TabPFN with smoothed features on all datasets. Note that TabPFN-v1 is limited to 10 classes (making evaluation impossible on datasets like Coauthor-CS with 15 classes), while NodePFN handles up to 20 classes. We have added these results to the revised paper in `Appendix H`.
>
> > Comparison between TabPFN with smoothed features and NodePFN on homophily datasets (accuracy %)
>
> | Dataset        | TabPFN-v1 (smoothed features) | NodePFN |
> |---|---|---|
> | AirBrazil      | 67.69                         | 75.38   |
> | AirEU          | 55.62                         | 57.00   |
> | AirUS          | 59.60                         | 61.66   |
> | Cora           | 74.06                         | 82.06   |
> | Citeseer       | 51.16                         | 67.30   |
> | Pubmed         | 75.96                         | 78.00   |
> | WikiCS         | 74.90                         | 75.98   |
> | Amazon-Photo   | 83.69                         | 90.53   |
> | Amazon-Comp    | 75.61                         | 81.42   |
> | DBLP           | 69.20                         | 74.71   |
> | Co-author CS   | N/A                           | 91.55   |
> | Co-author Physics | 87.93                      | 93.43   |
> | Deezer         | 48.17                         | 53.45   |
>
> > Comparison between TabPFN-v1 with smoothed features and NodePFN on heterophily datasets (accuracy %)
>
> | Dataset        | TabPFN-v1 (smoothed features) | NodePFN |
> |---|---|---|
> | Cornell        | 42.16                         | 71.89   |
> | Texas          | 56.22                         | 76.22   |
> | Wisconsin      | 51.37                         | 79.22   |
> | Chameleon      | 41.42                         | 50.13   |
> | Actor          | 25.29                         | 32.99   |
> | Minesweeper    | 80.07                         | 80.66   |
> | Tolokers       | 78.05                         | 78.61   |
> | Amazon-Ratings | 44.24                         | 44.68   |
> | Questions      | 97.02                         | 97.02   |
> | Squirrel       | 40.42                         | 43.40   |

---

> ### Author Response · Authors · 2025-11-22
>
> **Response to W4: Scalability Concern**
>
> Thank you for raising these limitations, which we explicitly acknowledge in `Section 6` and our `Conclusion`. We provide both theoretical complexity analysis and empirical runtime measurements to address scalability concerns.
>
> NodePFN has $O(L (n^2d + |E|d))$ complexity per forward pass, combining the attention branch $O(n^2d)$ and MPNN branch $O(|E|d)$. Critically, the quadratic attention complexity, inherited from the original PFN architecture, dominates total cost, while our added MPNN contributes minimally. This is confirmed by empirical measurements:
>
> | Ablation | Pre-training (per Epoch) | Inference Time (s, 23 datasets) |
> | -------- | -------- | -------- |
> | w/o MPNN | 12m 46s | 47.25s |
> | NodePFN (with MPNN) | 12m 55s | 47.78s |
>
> Adding the MPNN branch increases pre-training time by only 9 seconds per epoch and total inference time by 0.53 seconds across all 23 datasets. This shows that our architectural contribution imposes negligible additional computational cost.
>
> We acknowledge that the $O(n^2)$ attention complexity limits scalability to very large graphs. Our benchmarks range from small to medium-large (e.g., Questions has 48,921 nodes), with NodePFN completing inference on Questions in 18.03 seconds. While this shows practical for large-scale graph learning benchmarks, we agree that million-node graphs would require architectural modifications (e.g., graph sampling and linear attention) and represent an important direction for future work.
>
> ---
>
> **Response to minor W1: Why training on labels a fundamental limitation?**
>
> Thank you for pointing out the ambiguity in our abstract. You're correct that the phrasing "training on labeled nodes" is unclear. The fundamental limitation we meant to convey is not the need for labeled nodes per se, but rather that GNNs must be retrained from scratch for each new graph dataset. Unlike our NodePFN, which generalizes universally after one-time pre-training, traditional GNNs do not generalize across different graphs. The GNNs trained on Cora cannot be directly applied to CiteSeer without retraining.
> We have revised the `Abstract` to clarify.
>
> ---
>
> **Response to minor W2: Philosophy of prior fitting and LLM**
>
> Thank you for the feedback on our motivation in the `Introduction`.
> Our LLM analogy may be misleading, since the principles differ fundamentally. The PFNs perform Bayesian inference via synthetic priors while LLMs learn from natural language corpora.
>
> We have revised the `Introduction` to remove the LLM comparison and focus on the PFN paradigm.
>
> ---
>
> **Response to minor W3: What is graph heterogeneity? Graph and tasks are different concepts. Tasks determine labels**
>
> Our use of "graph heterogeneity" could be confused with heterophily or task diversity.
> We meant the structural diversity across different graph datasets, not heterogeneity within a single graph.
> We have updated the `Line 44`. Our updated sentence clarifies that we're discussing structural variations among graph datasets rather than using the overloaded term "heterogeneity".
> Thank you for helping us improve precision in our terminology.
>
> ---
>
> **Response to minor W4: Graphany does have designs for heterogeneity. In the aggregation stage, it has filter like ($I-A$)**
>
> You're correct that GraphAny includes heterophily-aware designs, such as ($I-A$) filters. Our statement was imprecise. GraphAny indeed handles heterophily but requires training separate models on different source datasets (GraphAny-Wisconsin vs GraphAny-Cora) to achieve optimal performance on target graphs. Our point is that while GraphAny addresses heterophily through architectural design, it still needs dataset-specific training because they have different version of GraphAny and their accuracy differ depend on the source dataset. We have revised our statement in `Introduction`.

---

> > ### Comment · Reviewer_f51f · 2025-11-24
> >
> > Thanks for the response. I have read several relevant submissions on prior-fitting GFM, and I think this one is of the highest quality. Despite the current version still suffering from potential scalability issues (no large-scale attempts), it would still be an interesting paper to present at the conference. I have changed my scores accordingly.

---

### Official Review · Reviewer_WYNZ · 2025-10-31

**Soundness:** 3
**Presentation:** 3
**Contribution:** 3
**Rating:** 6
**Confidence:** 4

**Summary:**

This paper introduces NodePFN, extending the Prior-Fitted Network (PFN) paradigm to graph neural networks. The key innovation is training a single model on thousands of synthetic graphs generated from carefully designed priors (random networks with controllable homophily and structural causal models) to learn posterior predictive distributions for node classification. This enables the model to perform inference on arbitrary real-world graphs without task-specific training, achieving 71.27% average accuracy across 23 benchmarks.

**Strengths:**

- The extension of PFNs to graphs represents a genuinely new direction. The idea of learning universal node classification patterns from synthetic priors rather than requiring graph-specific training is novel and addresses a fundamental limitation of current GNNs.
- The paper provides theoretical analysis (Theorems in Section 2.3) showing how graph structure affects node features' impact on GNN performance, motivating why synthetic priors with controlled properties can capture real-world patterns.
- Testing on 23 diverse benchmarks, including both homophilic and heterophilic graph,s demonstrates wide applicability. The 71.27% average accuracy with a single model is impressive, particularly the 65.14% on heterophilic graphs where traditional GNNs struggle.
- The combination of contextual SBMs with controllable homophily (0.1-0.9) and structural causal models for feature-label relationships shows thoughtful prior design that systematically covers graph diversity.
- The "train once, deploy everywhere" paradigm offers practical benefits, removing the need for dataset-specific training while maintaining competitive performance.

**Weaknesses:**

- The quadratic complexity of attention mechanisms and fixed constraints on class numbers (max 20) and feature dimensions significantly limit applicability to real-world large-scale graphs. This is acknowledged but not addressed.
- While the paper describes the priors used, it lacks ablation studies on prior design choices. Why these specific distributions? How sensitive is performance to prior specification? The connection between prior properties and downstream performance needs deeper investigation.
- The paper mentions ~250,000 synthetic graphs and 6 GPU hours of training, but doesn't provide detailed computational comparisons with training multiple task-specific GNNs. The amortization argument needs quantitative support.
- The paper primarily compares against basic GNNs (GCN, GAT) and GraphAny. Comparisons with more recent heterophily-specific methods (e.g., H2GCN, GPRGNN) and other universal graph methods would strengthen the evaluation.
- What determines when NodePFN will succeed or fail? The paper doesn't clearly characterize the conditions under which synthetic priors successfully capture real-world patterns or provide failure case analysis.
- The dual-branch architecture combining attention and message passing is described superficially. Ablations on architectural choices and their contributions are limited (only Table 3 provides some ablations).

**Questions:**

1. How does performance degrade as graphs exceed the training distribution in terms of size, feature dimensions, or class numbers? Can you provide scaling experiments?
2. Can you provide more extensive ablations on the synthetic prior design? How do different homophily distributions, SCM architectures, or graph generation models affect performance?
3. Why does NodePFN sometimes underperform GraphAny models trained on specific datasets (e.g., Amazon-Comp: 81.42% vs 83.04%)? What patterns are difficult to capture with synthetic priors?
4. Could you compare against more recent heterophily-specific GNN methods and explain when NodePFN's universal approach is preferable to specialized architectures?
5. Is there a way to adapt or fine-tune NodePFN for specific graphs while maintaining most of the universal knowledge? Maybe this could address performance gaps on specific datasets.

---

> ### Author Response · Authors · 2025-11-22
>
> We appreciate Reviewer WYNZ for the positive feedback and insightful comment,
>
> ---
>
> **Response to Q1: Scalability Ablation**
>
> Thank you for this question. We conducted scaling experiments by pre-training separate NodePFN models with different maximum feature dimensions (50, 100, 200) and maximum class numbers (5, 10, 20), creating distinct synthetic prior datasets for each configuration.
> For feature dimensions, the 100-dimensional setup achieves optimal performance across all datasets. Though performance remains competitive at 50d and 200d, demonstrating robustness to feature dimension variations.
> For class numbers, the 5-class maximum cannot evaluate Cora (which has 7 classes), while the 10-class setup still achieves strong performance on Wisconsin but limits dataset coverage.
> We therefore adopt 20 maximum classes for our main model to ensure comprehensive coverage of all 23 benchmark datasets.
>
> | # features    | Cora   | Wisconsin | Tolokers |
> |---|---|-----------|----------|
> | 50              | 81.50  | 80.43     | 78.38    |
> | 100           | 82.06 | 81.18 | 78.61|
> | 200             | 81.09  | 79.22   | 78.43    |
>
> | # classes    | Cora   | Wisconsin | Tolokers |
> |------|-----|-----|-------|
> | 5              | N/A  | 76.08    | 78.21    |
> | 10             | 79.12  | 80.43  | 76.78    |
> | 20 | 82.06 | 81.18 | 78.61 |
>
> ---
>
> **Response to Q2&W2: Synthetic prior design ablations**
>
> For homophily distribution, we tested various homophily ranges and graph generation models. Low-homophily priors (0.1-0.3) underperform on homophilic Cora, while high-homophily priors underperform on heterophilic Wisconsin (78.20%). Barabási-Albert graphs show inconsistent performance, suggesting power-law degree distributions alone are insufficient. The full homophily spectrum (0.1-0.9) is important for universal coverage.
>
> | Ablation              | Cora   | Wisconsin | Tolokers |
> |------|----|----|-----|
> | Only ER               | 80.62  | 80.39     | 77.18    |
> | Only cSBM             | 81.26  | 78.82     | 77.30    |
> | Only cSBM (low h) | 79.89  | 79.98     | 77.65    |
> | Only cSBM (high h)| 80.42  | 78.20     | 77.23    |
> | Only BA               | 74.18  | 80.57     | 74.63  |
> | **NodePFN**           | **82.06** | **81.18** | **78.61** |
>
>
> For ER/cSBM ratio analysis, we conducted ablation varying ratio.
> The balanced 50/50 ratio consistently achieves optimal or near-optimal performance across all homophily regimes.
>
> | ER% | Cora   | Wisconsin | Tolokers |
> |---|---|---|---|
> | 100% (Only ER) | 80.62  | 80.39 | 77.18 |
> | 80%     | 80.90  | 81.30 | 77.20 |
> | 50%   | 82.06 | 81.18 | 78.61 |
> | 20%     | 82.01  | 78.75 | 78.10 |
> | 0%  (Only cSBM) | 81.26 | 78.82 | 77.30 |
>
> The  50/50 ratio consistently achieves optimal or near-optimal performance across all homophily regimes. Wisconsin benefits most from higher ER ratios, likely because ER's unbiased topology provides crucial structural diversity for heterophilic graph. This shows that both ER's unbiased topology and cSBM's community structure contribute complementary inductive biases important.

---

> ### Author Response · Authors · 2025-11-22
>
> **Response to Q4&W4: Heterophily-specific GNNs & universal graph methods**
>
> We have conducted comparisons with heterophily-specific GNNs (H2GCN, GPRGNN, FAGCN) and added results in Appendix F. NodePFN achieves the best performance on 7 out of 9 datasets despite, while these specialized GNN methods require dataset-specific optimization. Importantly, we achieve improvements on Chameleon and Squirrel, demonstrating that our synthetic prior approach effectively captures heterophily patterns. On Texas and Actor where NodePFN slightly underperforms, the gap is marginal and stems from these specialized architectures' ability  specifically for each dataset's unique heterophily characteristics.
>
> | Dataset | Chameleon | Squirrel | Cornell | Texas | Actor | Wisconsin | Amazon-Ratings | Coauthor-CS | Coauthor-Physics |
> |---|---|---|---|---|---|---|---|---|---|
> | H2GCN | 41.07 ± 2.65 | 35.10 ± 1.15 | 65.77 ± 6.80 | **76.58 ± 1.56** | **35.86 ± 1.03** | 75.82 ± 1.13 | 40.87 ± 0.11 | 88.45 ± 0.97 | 92.86 ± 0.36 |
> | GPRGNN | 39.69 ± 1.15 | 38.95 ± 1.99 | 40.54 ± 2.01 | 65.77 ± 1.56 | 33.94 ± 0.95 | 75.21 ± 4.08 | 42.23 ± 0.25 | 91.49 ± 0.39 | 92.76 ± 0.20 |
> | FAGCN | 37.24 ± 3.54 | 36.78 ± 3.11 | 60.38 ± 1.82   | 68.44 ± 1.78 | 34.87 ± 1.25 | 72.02 ± 5.24 | 44.12 ± 0.31     | 91.07 ± 1.28 | 92.34 ± 0.40 |
> | **NodePFN** | **50.13 ± 3.30** | **43.40 ± 1.03**| **71.89 ± 2.76** | 76.22 ± 7.53 | 32.99 ± 1.09 | **79.22 ± 6.97**   | **44.68 ± 0.48** | **91.55 ± 0.32** | **93.43 ± 0.13** |
>
> Additionaly, we provide comprehensive comparisons using the same GLBench[5] splits for fair evaluation:
>
> | Method | Cora | Citeseer | Pubmed | WikiCS |
> | -------- | -------- | -------- | -------- | -------- |
> | InstructGLM     | 69.10 | 51.87 | 71.26 | 45.73 |
> | GraphText [2]   | 76.21 | 59.43 | 75.11 | 67.35 |
> | LLaGA [3]       | 74.42 | 55.73 | 68.82 | 73.88 |
> | OFA   [4]       | 75.24 | 73.04 | 75.61 | 77.34 |
> | NodePFN         | 76.38 | 63.08 | 69.05 | 76.29 |
>
> NodePFN achieves competitive or superior performance compared to LLM-based graph foundation models without requiring text descriptions or language model dependencies. While LLM-based methods leverage pre-trained language knowledge, NodePFN uses pre-trained patterns from massive synthetic prior data. This makes our approach complementary rather than competitive, suitable for graphs with numerical features where creating meaningful text descriptions is challenging or artificial. We added these comparisons in `Appendix H`.
>
> > [1] Ye, Ruosong, et al. "Language is All a Graph Needs." EACL (Findings). 2024.
> >
> > [2] Zhao, Jianan, et al. "GraphText: Graph Reasoning in Text Space." Adaptive Foundation Models: Evolving AI for Personalized and Efficient Learning.
> >
> > [3] Chen, Runjin, et al. "Llaga: Large language and graph assistant." arXiv preprint arXiv:2402.08170 (2024).
> >
> > [4] Liu, Hao, et al. "One For All: Towards Training One Graph Model For All Classification Tasks." The Twelfth International Conference on Learning Representations.
> >
> > [5] Li, Yuhan, et al. "Glbench: A comprehensive benchmark for graph with large language models." Advances in Neural Information Processing Systems 37 (2024): 42349-42368.
>
> ---
>
> **Response to Q3 & W5: Success/failure conditions**
>
> PFN's attention mechanism treats nodes as independent context elements, failing to directly encode degree or connectivity patterns. We introduced a basic GCN-based MPNN branch to capture graph structure, but this simple message-passing design shows limitations on extreme power-law graphs with very high-degree hubs (e.g., Amazon-Computer).
>
> NodePFN outperforms GraphAny on 3 of 4 power-law graphs despite using only basic GCN aggregation and not explicitly modeling power-law distributions. The Amazon-Computer gap is minor, while NodePFN achieves highest average accuracy across all 23 benchmarks.
>
> | Dataset | Power-law $\alpha$ | NodePFN | GraphAny (Wisconsin) |
> | ------- | -------- | -------- | ----- |
> | Amazon-ratings | 3.6059 | 44.68 | 42.57 |
> | WikiCS  |  3.5788 | 75.98 | 73.77 |
> | Amazon-Photo | 2.9223 | 90.53 | 90.18 |
> | Amazon-Computer  | 2.8389 | 81.42 | 82.00 |
>
> On high-clustering networks (top-3 hightest clustering coefficient), NodePFN ranks 1st on Air-Brazil and Chameleon, validating that cSBM's implicit community structure and triangle generation effectively capture motif patterns without explicit modeling. But. on Amazon-ratings, NodePFN does not outperform GCN.
> GCN's superior performance stems from its transductive setting, well-optimized use of unobserved validation sets.
>
> | Dataset | Clustering Coeff. | NodePFN Accuracy | Rank |
> | ------- | -------- | ------- | -------- |
> | Air-Brazil | 0.6364 | 75.38 | 1/8 |
> | Amazon-ratings | 0.5816 | 44.68 | 4/8 |
> | Chameleon  |  0.5769 | 50.13 | 1/8 |

---

> ### Author Response · Authors · 2025-11-22
>
> **Response to Q5: Fine-tuning**
>
> Thank you for your suggestion. We conducted preliminary fine-tuning experiments. For example, on Wisconsin, fine-tuning improved performance from 79.22±6.97 to 80.39±4.30. In Texas, the fine-tuned NodePFN improved from 76.22±7.53 to 78.38±5.13.
>
> ---
>
> **Response to W1: Limitations**
>
> Thank you for highlighting these limitations, which we explicitly acknowledge in Section 6 and our Conclusion. The quadratic attention complexity is beyond our current scope and represents a separate future research direction. However, the feature dimension constraint is already addressed through truncated SVD, which successfully handles our benchmarks ranging from 7 to 8,415 features. The class limit of 20 covers all our 23 benchmarks (maximum 15 classes) and most real-world node classification tasks.
>
> ---
>
> **Response to W3: Computational cost**
>
> We have added quantitative comparisons in `Appendix K` to provide runtime.
>
> | Method | GCN | NodePFN
> | -------- | -------- | -------- |
> | Avg. Accuracy (%) | 66.63  | 71.27 |
> | Avg. Ranking | 5.86 | 1.70 |
> | Total Training Time (s) | 188 | - |
> | Total Inference Time (s) | 12.35 | 47.78 |
>
> As shown in the table, GCN requires 200.35 seconds total (188s training + 12.35s inference) on all 23 datasets with fixed hyperparameters. If we assume that model tuning requires 5-10 trials, this will significantly increase the cost. NodePFN requires only 47.78 seconds (inference only).
>
> ---
>
>
> **Response to W6: Ablation study of architecture**
>
> While Table 3 in our paper already includes ablations on sequential processing (NodePFN-Seq) and model capacity (NodePFN-L6), we have now added a missing ablation: NodePFN w/o MPNN (Transformer-only architecture). This new ablation, presented in `Appendix J`, demonstrates the MPNN branch's contribution to the dual-branch design.
>
> The results reveal that removing the MPNN branch causes  performance degradation. The reason is that local neighborhood structure carries critical signals that pure attention mechanisms does not capture sufficiently.
>
> | Ablation      | Cora   | Wisconsin | Tolokers |
> |---------------|--------|-----------|----------|
> | NodePFN-L6    | 53.10  | 72.94     | 78.00    |
> | NodePFN-Seq   | 80.64  | 78.82     | 77.88    |
> | NodePFN w/o MPNN | 75.50  | 70.10     | 78.09    |
> | **NodePFN**   | **82.06** | **81.18** | **78.61** |

---

### Official Review · Reviewer_ogba · 2025-10-31

**Soundness:** 3
**Presentation:** 4
**Contribution:** 3
**Rating:** 8
**Confidence:** 4

**Summary:**

The authors propose NodePFN, a foundation model for node classification on arbitrary graphs. Following recent results on foundation models for tabular data, whose main insight is training models on synthetically generated data that resembles relationships present in real-world data, the authors propose to train a GNN on synthetically generated graphs. In order to accommodate for the variable number of node features and labels across different datasets, as well as to achieve in-context learning for new graphs, the authors propose an architecture based on self-attention. Experiments indicate that NodePFN effectively generalizes to real world datasets, surpasing the performance of supervised methods that require data-specific training.

**Strengths:**

1. The paper addresses an important limitation of supervised methods that require training from scratch, or at best fine-tuning for every new dataset, thus reducing the cost of deploying predictive models on graphs.
2. The paper introduces well-motivated synthetic data generation methods that cover diverse network topologies, and an architecture for learning in context intended to be pre-trained with synthetic data.
3. The experiments are comprehensive, demonstrating the effectiveness of NodePFN agains a broad range of baselines and in ablation experiments.

**Weaknesses:**

1. The novelty of the work is slightly limited, as it is a natural extension of well-known results from TabPFN and its follow-up works, to the graph domain.
2. Important details about the generated data are missing. How many graphs in total were generated? What is the number of nodes and edges in them?
3. NodePFN looks like a costly model for training and inference. Some indication of training cost is given in the appendix ("6 GPU hours"), but a formal derivation of complexity is missing from the paper. This is especially important considering that the method requires a new architecture that differs from established ones like the GCN.
4. Important details about the architecture are not clear. Can NodePFN handle node features, especially a variable number? Section 6 states that this is a limitation, requiring a maximum number of dimensions, but it is not clear how this is dealt with. Does it involve some form of padding along the feature dimension?
5. A sometimes implicit yet strong motivation for works such as TabPFN, and in this case NodePFN, is that generating a collection of datasets containing complex relations between variables that is large and random enough allows pretraining models that generalize to real data. The question here is: how do we know what is large and random enough to guarantee generalization? Is NodePFN good because our knowledge about the datasets in Table 1 allows us to generate data like them? And if so, how robust is it to distribution shift?
6. The architecture can in principle be fine-tuned but these experiments are not considered in the paper. It would be interesting to know how much of an improvement this can bring.

**Questions:**

1. Can you please provide further information about the statistics of the generated datasets?
2. The appendix states that NodePFN is trained with one synthetic graph per epoch, and that a batch size of 8 is used. What does a "batch" mean in this context? In the simplest setting, training a GNN requires using the full graph (i.e. the complete feature and adjacency matrices) in order to do an unbiased forward pass.
3. Can you please clarify how NodePFN handles input features whose number varies across datasets?
4. Do you have any thoughts on the generalization properties of NodePFN and its robustness to distribution shift? It would be interesting to know how confident you are that it will work well for every graph in the wild.

---

> ### Author Response · Authors · 2025-11-22
>
> We sincerely thank Reviewer ogba for the thoughtful feedback.
>
> ---
>
> **Response to Q1&W2: Statistics of prior data**
>
> Thank you for your feedback. Sorry for omitting the detailed explanation earlier. During pretraining, we synthesize and generate about 250,000 different graphs over a total of 30 epochs. The number of nodes is fixed at 1024. On average, each graph contains 8.79 classes and 12,706.4 edges. Please refer to `Appendix J` for the edge distributions and the number of classes for each graph.
>
> ---
>
> **Response to Q2: Meaning of batch**
>
> Thank you for this question. The description is incorrect, and we apologize for the confusion caused by our phrasing.
>
> In our prior data-fitting training context, a "batch" refers to a set of complete, independently generated synthetic graph datasets that are processed in parallel for a single gradient update.
>
> Our batch size of 8 means that 8 new synthetic graphs are generated on-the-fly for every training step. Therefore, our original phrase that `"one synthetic graph was generated for each epoch"` was incorrect. An "epoch" is a unit of 1,024 training steps, meaning that a total of 8,192 unique synthetic graphs (1,024 steps × 8 graphs/step) were generated and used within each epoch. We revised the paper to clarify this procedure.
>
>
> **Response to Q3&W4: Handling a number of features**
>
> To clarify, our NodePFN is designed to handle a variable number of input features, up to a predefined maximum, similar to TabPFN. This is achieved via a flexible encoder mechanism. We add the details of this encoder for handling the features in `Appendix B.4`. When a graph dataset has fewer features than the maximum number of features, we apply zero-padding to the feature dimension to match the required input size of our NodePFN.
>
> Conversely, for cases where the feature dimension exceeds the maximum, we apply a standard dimensionality reduction technique before feeding the data to the model. Specifically, we use truncated SVD to project the features down to the required input dimension of NodePFN.
>
> ---
>
> **Response to Q4&W5: Generalization**
>
> We acknowledge that coverage of all possible graphs is impossible, but our approach provides generalization through two key mechanisms. First, our many synthetic graphs, systematically varied in homophily, structure, and feature-label relationships, provide a broad prior that covers fundamental graph patterns. Second, NodePFN's in-context learning uses the target graph's labeled nodes as context, performing posterior inference without fine-tuning.
> Our generalization and robustness to distribution shift come from this meta-learning approach. Even when encountering unseen graphs, the model uses its learned inference strategies to generalize.

---

> ### Author Response · Authors · 2025-11-22
>
> **Response to W1: Novelty concerns**
>
> While we build upon TabPFN's paradigm, our contribution represents a fundamental reconceptualization rather than natural extension.
>
> This work requires a design to reflect the structural dependencies and generation mechanisms unique to graphs. In particular, integrating MPNN to handle inter-node relationships and introducing priors based on cSBM and ER random networks are graph-specific elements not present in existing PFN series. This configuration goes beyond simple application level, reconstructing in a form suitable for graph domains while maintaining PFN philosophy, and we believe the main contribution of this research is being the first to systematically extend the PFN approach in the graph domain. It required significant effort for prior data design choices, and as shown in the analysis of the synthetic prior design provided below and `Appendix I` we were able to arrive at our design.
>
> | Ablation              | Cora   | Wisconsin | Tolokers |
> |---|---|---|---|
> | Only ER               | 80.62  | 80.39     | 77.18    |
> | Only cSBM             | 81.26  | 78.82     | 77.30    |
> | Only cSBM (low $h$ 0.1 to 0.3) | 79.89  | 79.98     | 77.65   |
> | Only cSBM (high $h$ 0.7 to 0.9) | 80.42  | 78.20     | 77.23   |
> | NodePFN           | 82.06 | 81.18 | 78.61 |
>
> ---
>
> **Response to W3: Complexity analysis**
>
> We agree that formal complexity analysis was missing and have now added it to `Appendix K`.
>
> While NodePFN requires one-time pre-training (6 GPU hours on 200K+ synthetic graphs), this enables direct inference on unlimited real-world datasets without gradient updates. Traditional GNNs such as GCN and GAT require separate training for each dataset. The training cost of GNNs accumulates with each new graph. For typical benchmarks, NodePFN inference completes in seconds.
>
> We also provide runtime on all 23 benchmarks we used. GCN requires 200.35 seconds total (188s training + 12.35s inference) across all 23 datasets with fixed hyperparameters. If we assume that model tuning requires 5-10 trials, this will significantly increase the cost. NodePFN requires only 47.78 seconds (inference only).
>
> | Method | GCN | NodePFN
> | -------- | -------- | -------- |
> | Avg. Accuracy (%) | 66.63  | 71.27 |
> | Avg. Ranking | 5.86 | 1.70 |
> | Total Training Time (s) | 188 | - |
> | Total Inference Time (s) | 12.35 | 47.78 |
>
> ---
>
> **Response to W6: Fine-tuning**
>
> Thank you for your interesting suggestion. We conducted preliminary fine-tuning experiments.
> For example, on Wisconsin, fine-tuning improved performance from 79.22±6.97 to 80.39±4.30. In Texas, the fine-tuned NodePFN improved from 76.22±7.53 to 78.38±5.13.

---

> > ### Comment · Reviewer_ogba · 2025-11-26
> >
> > Thank you for the additional details. I think these are very valuable and I would encourage you to include them in the final version of the paper.
> >
> > My positive assessment of the paper remains so I will maintain my score.

---

### Author Response · Authors · 2025-12-02
**Summary of Rebuttal Discussion for New Area Chair**

Dear new AC and Reviewers

We thank the new AC for your time and contribution to the community, and we are grateful to all reviewers for their thoughtful suggestions and kind recognition of our contribution.

Contributions recognized by reviewers include:
- Novel "train once, deploy everywhere" paradigm removing dataset-specific training requirements
- Well-designed synthetic prior generation using contextual SBMs with controllable homophily and structural causal models that systematically cover graph diversity
- Strong performance across 23 benchmarks, especially on heterophilic graphs
- Comprehensive experimental validation with extensive ablation studies

---

## Reviewer Score Changes

During the discussion period, reviewer scores changed from `8, 6, 4, 4` to `8, 6, 6, 6` before being rolled back due to the OpenReview anonymity leak.

| Reviewer | Initial Rating | Rating after Our Response |
|---|---|---|
| `ogba` | 8 | 8 |
| `WYNZ` | 6 | 6 |
| `f51f` | 4 | **6** |
| `M3ir` | 4 | **6** |

> **Summary of Reviewer Responses:**
>
> - Reviewer `ogba`: Maintained their positive score (8) and commented that our additions on prior data statistics, synthesis-method comparisons, and complexity analysis are "very valuable" for the final version.
>
> - Reviewer `WYNZ`: Did not respond further during the discussion period despite our experimental responses on prior data, heterophily-specific GNNs, and LLM-based foundation model comparisons.
>
> - Reviewer `f51f`: **Increased their score from 4 to 6** after reviewing our theoretical discussion and additional baseline experiments, noting this work is "of the highest quality" among prior-fitting GFM submissions.
>
> - Reviewer `M3ir`: **Increased their score from 4 to 6**, stating our response "effectively addresses my concerns" regarding heterophily-specific GNN baselines, synthetic graph prior analysis, and computational analysis.

Please refer to the detailed summary below for specific concerns addressed and experiments conducted for each reviewer.

---

## Summary of Reviewer Concerns and Our Responses


| Concerns & Suggestions | By | Our Response |
|------------------------|-----|--------------|
| Statistics of synthetic prior data | `ogba` | We provided detailed statistics (Appendix J). |
| Feature handling mechanism | `ogba` | We clarified flexible encoder with zero-padding and truncated SVD (Appendix B.4). |
| Novelty concerns vs TabPFN | `ogba` | We clarified graph-specific contributions: MPNN integration, cSBM/ER priors with extensive ablations. |
| Fine-tuning potential | `ogba`, `WYNZ` | We conducted preliminary fine-tuning experiments that showed performance improvements. |
| Computational complexity | `ogba`, `WYNZ`, `f51f`, `M3ir` | We added formal complexity analysis and runtime comparisons (Appendix K). |
| Synthetic prior design | `ogba`, `WYNZ`, `M3ir` | We conducted systematic ablations on homophily ranges and ER/cSBM ratios (Appendix I). |
| Heterophily-specific GNNs | `WYNZ`, `M3ir` | We compared H2GCN, GPRGNN, FAGCN; NodePFN wins 7/9 datasets (Appendix F). |
| Universal graph methods | `WYNZ` | We compared LLM-based foundation models (InstructGLM, GraphText, LLaGA, OFA) using GLBench datasets. |
| Architectural ablation | `WYNZ` | We added w/o MPNN ablation in Appendix J. |
| Theoretical justification | `f51f` | We added a theoretical discussion demonstrating that NodePFN preserves Müller et al. (2022)'s guarantees (Appendix E). |
| TabPFN with smoothed features | `f51f` | We conducted a comprehensive comparison; NodePFN outperforms on all 23 datasets (Appendix H). |
| Method description clarity | `f51f` | We refined Section 3.3 and added Appendix B.5. |
| Structural graph properties | `M3ir` | We analyzed graph structural properties (power-law distributions, clustering coefficients) and demonstrated strong performance on benchmarks exhibiting these characteristics |
| Temporal graphs | `M3ir` | We conducted dynamic experiments showing stable performance. |

---

Sincerely,

The Authors of Submission24880

---

### Public Comment · ~Dmitry_Eremeev1 · 2026-03-07

Dear authors,

Thank you for your great work!

I have a question regarding the results reported in Table 1. Could you please clarify the exact methodology used to compute the aggregated metrics (average accuracy and average rank)?

I attempted to reproduce these aggregated metrics based on the individual dataset scores provided in the table. I noticed that the re-computed aggregations systematically differ from the reported values. To give a few specific examples using the homophilic datasets subset:

1. Average Accuracy: For MLP, the paper reports 56.43, whereas computing the average of the column gives 54.81.
2. Average Rank: When computing ranks per dataset (using `df.rank(axis=1, ascending=False)` in `pandas`), I find that GAT achieves an average rank of 2.62 and NodePFN achieves 2.69. However, the paper reports an average rank of 4.54 for GAT, compared to 1.69 for NodePFN.

Since the differences seem consistent across the table, I’m wondering whether the aggregated metrics were computed using a different procedure than a plain unweighted mean/ranking?

For transparency, here is the script I used to re-compute the aggregations: https://gist.github.com/eremeev-d/d74c1a5eff266809b57dfe1b173172c5.

Best regards,

Dmitry

---

### Meta-Review · Area_Chair_SBh7 · 2026-01-08

**Summary:**

This paper introduces NodePFN, a node classification framework that generalizes to arbitrary graphs without graph-specific training. The method learns posterior predictive distributions through large-scale pretraining on synthetic graphs with controllable homophily and structural properties, and combines context–query attention with local message passing to enable graph-aware in-context learning. The paper targets an important open problem in graph learning: achieving strong zero-shot generalization across diverse graph structures without per-dataset retraining.

During the rebuttal period, the authors provided a very detailed response to the reviews. They added extensive ablations on synthetic prior design (including homophily ranges and ER/cSBM mixtures), complexity and runtime analyses, comparisons to heterophily-specific GNNs (H2GCN, GPRGNN, FAGCN), and evaluations against LLM-based graph foundation models. Additional experiments on fine-tuning, temporal graphs, and structural graph properties further strengthened their evaluations. As a result, reviewer scores improved substantially. Two reviewers (f51f and M3ir) explicitly raised their scores after reviewing the authors’ detailed responses and additional experiments. Reviewer ogba maintained a strong positive score throughout, emphasizing that the added analyses on synthetic prior statistics, feature handling, and complexity were “very valuable.” Reviewer WYNZ did not follow up during discussion but it seems that their original concerns were addressed through new ablations and comparisons.

Overall, the paper makes a strong contribution to graph representation learning by demonstrating that synthetic data pretraining can yield robust zero-shot performance across a wide range of graph regimes. The contribution is timely, the responses were constructive and effective, and reviewers converged toward acceptance. I believe the work is technically solid, well-motivated, and of clear interest to the community, and it merits acceptance.

**Reviewer Concerns:**

There were no major reviewer concerns outstanding.

**Reviewer Scores:**

Both Reviewers f51f and M3ir indicated that they would raise their scores to 6 in favor of acceptance.

---

### Decision · Program_Chairs · 2026-01-26

Accept (Poster)